# Spatially resolved cell atlas of the mouse primary motor cortex by MERFISH

Meng Zhang[1,2,3], Stephen W. Eichhorn[1,2,3], Brian Zingg[4,6], Zizhen Yao[5], Kaelan Cotter[4,6], Hongkui Zeng[5], Hongwei Dong[4,6] & Xiaowei Zhuang[1,2,3 ✉]

A mammalian brain is composed of numerous cell types organized in an intricate manner to form functional neural circuits. Single-cell RNA sequencing allows systematic identification of cell types based on their gene expression profiles and has revealed many distinct cell populations in the brain[1,2]. Single-cell epigenomic profiling[3,4] further provides information on gene-regulatory signatures of different cell types. Understanding how different cell types contribute to brain function, however, requires knowledge of their spatial organization and connectivity, which is not preserved in sequencing-based methods that involve cell dissociation. Here we used a single-cell transcriptome-imaging method, multiplexed error-robust fluorescence in situ hybridization (MERFISH)[5], to generate a molecularly defined and spatially resolved cell atlas of the mouse primary motor cortex. We profiled approximately 300,000 cells in the mouse primary motor cortex and its adjacent areas, identified 95 neuronal and non-neuronal cell clusters, and revealed a complex spatial map in which not only excitatory but also most inhibitory neuronal clusters adopted laminar organizations. Intratelencephalic neurons formed a largely continuous gradient along the cortical depth axis, in which the gene expression of individual cells correlated with their cortical depths. Furthermore, we integrated MERFISH with retrograde labelling to probe projection targets of neurons of the mouse primary motor cortex and found that their cortical projections formed a complex network in which individual neuronal clusters project to multiple target regions and individual target regions receive inputs from multiple neuronal clusters.

The mammalian cerebral cortex is a highly organized structure that supports sensory, motor and cognitive functions. Classification of neuronal cell types is central to deciphering cortical circuits[2,6]. The glutamatergic neurons in the cortex are classified by their projection properties into, for example, intratelencephalic (IT) neurons, subcerebral projection neurons (or pyramidal tract neurons) and corticothalamic (CT) projection neurons[7]. The GABAergic neurons can be classified based on their developmental origin into neurons derived from the medial ganglionic eminence and the caudal ganglionic eminence, which can be further classified by marker genes such as parvalbumin (*Pvalb*), somatostatin (*Sst*), vasoactive intestinal polypeptide (*Vip*) and *Lamp5* (ref. [8]). Recent single-cell transcriptomics studies have revealed a high diversity of cells in the brain[9–11] and reported dozens to about a hundred cell types within individual cortical regions[12–14]. However, a high-resolution map of the spatial organization and connectivity of different cell types in the cortex is still lacking.

Recently, several spatially resolved transcriptomics methods have been developed, including both imaging-based transcriptomics methods with single-cell resolution[15,16] and methods based on spatially resolved RNA capture followed by sequencing[17]. Among these,

MERFISH is a single-cell genome-scale imaging method, which massively multiplexes single-molecule fluorescence in situ hybridization (FISH)[18,19] using error-robust barcoding, combinatorial labelling and sequential imaging and allows simultaneous imaging of more than 10,000 genes in individual cells[5,20]. MERFISH allows in situ identification and spatial mapping of cell types in complex tissues, including the brain[21]. Here we used MERFISH to identify distinct cell populations and map their spatial organization in the mouse primary motor cortex (MOp). By integrating MERFISH with retrograde labelling, we further revealed the complexity of projection patterns of these molecularly defined cell types.

## MERFISH imaging and cell classification

We selected a panel of 258 genes for MERFISH imaging, including canonical marker genes for major neuronal and non-neuronal cell types in the cortex selected based on previous knowledge[12,13,21], as well as marker genes selected based on differential gene expression and mutual information[22] analyses of neuronal clusters identified by a companion single-cell and single-nucleus RNA sequencing (scRNA-seq and

[1]Howard Hughes Medical Institute, Harvard University, Cambridge, MA, USA. [2]Department of Chemistry and Chemical Biology, Harvard University, Cambridge, MA, USA. [3]Department of Physics, Harvard University, Cambridge, MA, USA. [4]Center for Integrative Connectomics, Mark and Mary Stevens Neuroimaging and Informatics Institute, Keck School of Medicine of USC, University of Southern California, Los Angeles, CA, USA. [5]Allen Institute for Brain Science, Seattle, WA, USA. [6]Present address: UCLA Brain Research & Artificial Intelligence Nexus, Department of Neurobiology, David Geffen School of Medicine, University of California, Los Angeles, CA, USA. ✉e-mail: zhuang@chemistry.harvard.edu

snRNA-seq, respectively) study[23] (Methods). We performed MERFISH measurements on a series of coronal slices at approximately 100-μm intervals along the anterior–posterior axis encompassing the MOp (Bregma +2.5 to −0.8). Individual RNA molecules were identified and assigned to individual cells (Extended Data Fig. 1a, b). Four of the 258 genes showed poor staining and were not included in subsequent analyses. The mean copy number per cell for individual genes obtained from MERFISH was reproducible between replicate mice and exhibited high correlation with the gene expression level measured by bulk RNA-seq (Extended Data Fig. 1c, d).

In total, we imaged and segmented approximately 300,000 individual cells in the MOp and its adjacent areas from two adult mice. Unsupervised clustering analysis[24,25] of the MERFISH-derived single-cell expression profiles identified 39 excitatory neuronal clusters, 42 inhibitory neuronal clusters and 14 non-neuronal clusters (Extended Data Fig. 2a), as well as 4 clusters exclusively outside the MOp (in the striatum or lateral ventricle), which were not included in subsequent analyses.

The MOp cell taxonomy showed a hierarchical organization (Extended Data Fig. 2a), with the first level separating glutamatergic, GABAergic and non-neuronal cell classes. The GABAergic class consisted of the neurons derived from the medial ganglionic eminence and the caudal ganglionic eminence, which were further divided into five subclasses based on their marker genes: *Pvalb*, *Sst*, *Vip*, *Sncg* and *Lamp5*. The glutamatergic neurons were classified into the following subclasses with distinct projection properties (identified on the basis of known marker genes[13]): layer 5 extratelencephalic projecting (L5 ET; also known as L5 pyramidal tract) neurons, layer 5/6 near-projecting (L5/6 NP) neurons, layer 6 CT (L6 CT) neurons, layer 6b (L6b) neurons and IT neurons. IT neurons were further classified into several subclasses (L2/3 IT, L4/5 IT, L5 IT and L6 IT, plus a distinct L6 IT *Car3* type) primarily based on layer-specific marker genes, additionally using the correspondence between MERFISH clusters and clusters identified by scRNA-seq and snRNA-seq for L4/5 and L5 IT classification (Methods). We also identified major non-neuronal cell subclasses, including oligodendrocytes, oligodendrocyte precursor cells, astrocytes, vascular leptomeningeal cells, microglia, perivascular macrophages, endothelial cells, smooth muscle cells and pericytes, based on marker genes. The subclasses determined by MERFISH (Extended Data Fig. 2b) showed excellent correspondence to those determined using scRNA-seq and snRNA-seq in the companion paper[23] (Extended Data Fig. 2c).

The 23 subclasses of cells contained a total of 95 clusters, for which we used nomenclature style of adding a numerical index following the subclass name (Extended Data Fig. 2a). The clusters identified by MERFISH also showed good correspondence to the clusters identified by scRNA-seq and snRNA-seq[23] (Extended Data Fig. 2d).

We imaged the MOp and its adjacent parts of the secondary motor (MOs) and primary somatosensory (SSp) areas, as well as other neighbouring regions. We registered the MERFISH images to the Allen mouse brain Common Coordinate Framework version 3 (CCF v3)[26] using the DAPI stains in our images and the Nissl stain in the Allen Reference Atlas[27] (Extended Data Fig. 2e and Methods), which allowed us to quantitatively determine the composition of cells both in the entire image region and in the MOp (Extended Data Fig. 2f).

## Spatial organization of cell types

MERFISH images provided a direct measurement of the spatial organization of transcriptomically distinct cell populations in the MOp and its adjacent areas (Fig. 1a). The layered organization of the glutamatergic subclasses, especially the IT subclasses, led to a laminar appearance for the overall cellular organization (Fig. 1a–c). Unlike the IT cells, which spanned across nearly all cortical layers, the ET, NP, CT and L6b cells populated only deeper layers (Fig. 1b, c). Individual glutamatergic clusters adopted spatially distinct, partially overlapping distributions along the direction of the cortical depth, and many of these clusters

assumed narrow distributions with widths smaller than the thicknesses of individual cortical layers.

The GABAergic neurons also showed a high level of spatial diversity. The *Lamp5*, *Sncg* and *Vip* subclasses were more populated in the upper layers, whereas the *Sst* and *Pvalb* subclasses were more abundant in deep layers (Fig. 1b, d), consistent with previous findings[28,29]. Notably, at the cluster level, most GABAergic clusters showed laminar distributions and preferentially reside within one or two cortical layers (Fig. 1d).

We also observed similar cortical depth distributions of neuronal clusters when we limited the analysis to the MOp region (Extended Data Fig. 3a) or an approximate upper limb region of the MOp[30,31] (Extended Data Fig. 3b). The distributions along the cortical depth exhibited small shifts between medial and lateral segments of the MOp or the upper limb region of the MOp for most neuronal clusters, with a few exceptions (such as L4/5 IT SSp 1 and 2, L6 IT *Car3*, L5 ET 4 and L6 CT 8) that showed brain region-dependent presence (Extended Data Fig. 3c). For example, L4/5 IT SSp 1 and 2 were primarily present on the lateral side of the MOp, extending from the SSp region, and L6 CT 8 was only present on the medial side of the MOp. Along the anterior–posterior direction, many neuronal clusters adopted broad distributions, whereas some were restricted to a relatively narrow anterior–posterior range (Extended Data Fig. 4).

We also mapped the spatial organizations of the non-neuronal cells (Fig. 1b, e). Among the three astrocyte clusters, astrocyte 1 exhibited a dispersed distribution across all layers, astrocyte 2 showed enrichment in L1 and the white matter, and astrocyte 3 was found almost exclusively in the white matter. The oligodendrocyte lineage was divided into oligodendrocyte precursor cells and three mature oligodendrocyte clusters, with the mature oligodendrocytes enriched in the white matter and the oligodendrocyte precursor cells distributed evenly across all layers. The vascular leptomeningeal cells formed the outermost layer of cells of the cortex. The other non-neuronal cell types exhibited more dispersed distributions across the cortical layers and white matter.

We noticed substantial spatial intermixing of different cell populations. To quantify the complexity of the cell composition in the neighbourhood of each cell, we determined the number of distinct cell clusters that were present in the neighbourhood of each cell and observed a high level of local cellular heterogeneity (Extended Data Fig. 5). The composition complexity of the cell neighbourhood increased towards deeper layers (Fig. 1f).

## L5 ET, L5/6 NP, L6 CT and L6b neurons

Transcriptomically, the L5 ET, L5/6 NP, L6 CT and L6b subclasses of neurons appeared as discrete cell populations, and each subclass was subdivided into finer clusters with more continuously varying gene expression profiles (Fig. 2a, Extended Data Fig. 6a, b).

Spatially, the L5 ET clusters were segregated into two sublayers with the L5 ET 1–3 clusters intermixed and distributed above L5 ET 5 (Fig. 2b). L5 ET 4 was largely absent from the MOp and only began to extend into the MOp on the ventral–lateral side in the anterior slices. It has been reported that two distinct L5 ET populations in upper and lower L5 of the anterior lateral motor cortex project to the thalamus and the medulla, respectively, and have specialized roles in motor control[32]. The L5 ET 5 cluster identified here by MERFISH corresponded to the L5 ET_1 cluster identified by single-cell transcriptomic and epigenomic data[23] (Extended Data Fig. 6b), which in turn corresponded to the medulla-projecting ET cluster identified by epi-retro-seq in a companion paper[33]. These results thus suggest that L5 ET 5 is a type of medulla-projecting neuron. MERFISH data further showed that L5 ET 5 was more enriched, and exhibited more pronounced spatial separation from the L5 ET 1–3 clusters, in the MOp than in the SSp (Extended Data Fig. 6c). Consistent with our finding, medulla-projecting ET neurons were also observed to be more abundant in the MOp than in the SSp by single-neuron reconstruction in another companion paper[34].

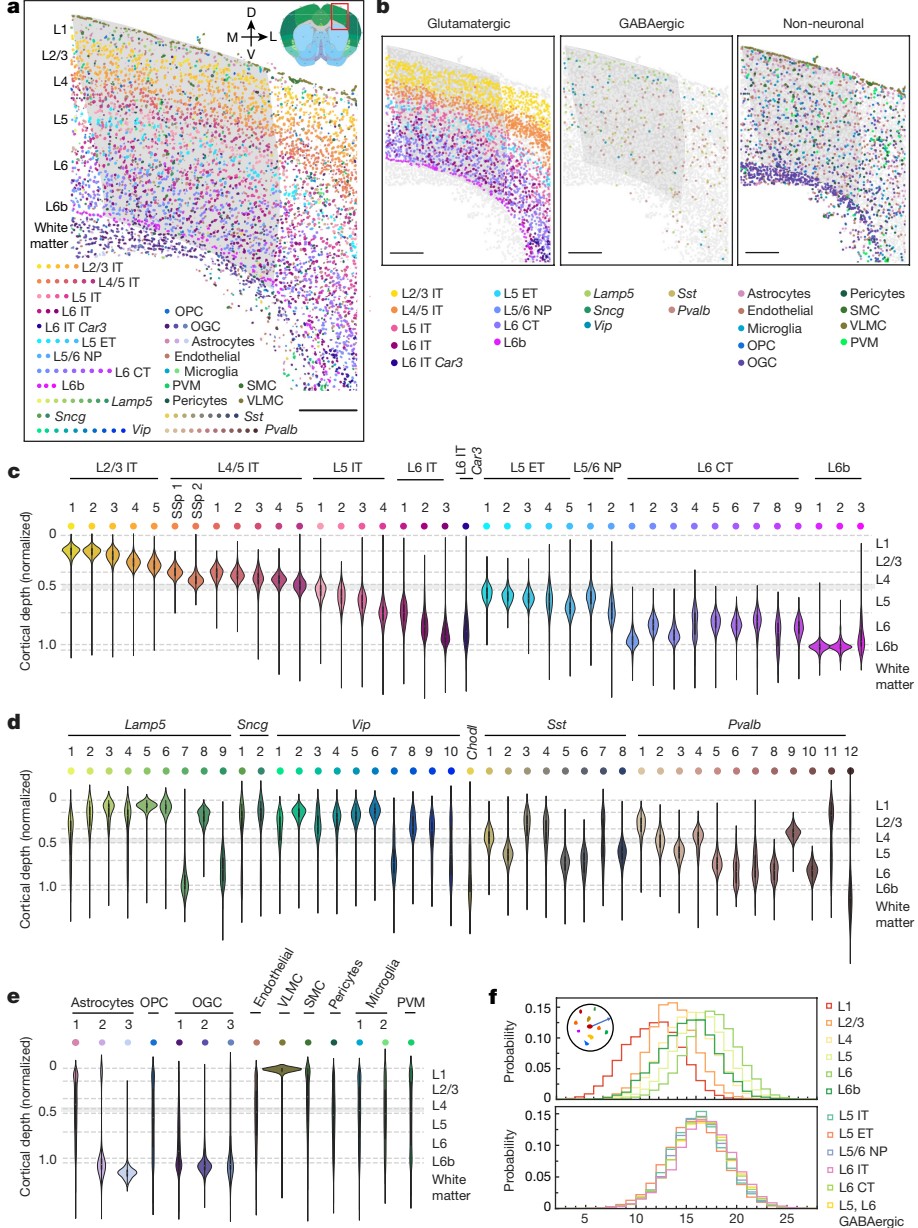

**Fig. 1 | Spatial organization of cells in the MOp and adjacent areas. a**, Spatial map of the cell clusters in a coronal slice (Bregma approximately +0.9). Cells are coloured by their cluster identities. The MOp region, determined based on the Allen CCF v3, is shaded in grey. The inset shows the brain region annotations in the Allen CCF v3 (http://atlas.brain-map.org/; credit: Allen Institute). D, dorsal; L, lateral; M, medial; OGC, oligodendrocyte; OPC, oligodendrocyte precursor cell; PVM, perivascular macrophage; SMC, smooth muscle cell; V, ventral; VLMC, vascular leptomeningeal cell. Scale bar, 400 μm. **b**, Spatial map of the cell subclasses in glutamatergic (left), GABAergic (middle) and non-neuronal (right) cells in the same slice as in **a**. Cells are shown as circles, with indicated cells coloured by subclasses and others in grey. Scale bars, 400 μm. **c**–**e**, The cortical depth distributions of the glutamatergic (**c**), GABAergic (**d**) and non-neuronal (**e**) clusters for the entire imaged region including the MOp and adjacent areas shown in the violin plots. The cortical depth of a cell is normalized by the cortical thickness in each slice, with 0 representing the cortical surface and 1 representing the median depth of the L6b cells. The dashed lines mark the layer boundaries, and the grey area marks an uncertainty range for the upper boundary of L5 (Methods). In the violin plots, the centre dot represents the median, the thick black bar represents the interquartile range, and the edges define minima and maxima. **f**, Probability distributions of the neighbourhood complexity of cells in each cortical layer (top) and in different cell subclasses in L5 and L6 (bottom). The neighbourhood complexity of a cell is defined as the number of different cell clusters present within a neighbourhood of 100 μm in radius surrounding the given cell.

The corticospinal projection neurons are also probably contained in the L5 ET subclass[35].

The L5/6 NP neurons were divided into two clusters (Fig. 2a), with L5/6 NP 1 mainly in L5 and L5/6 NP 2 extending into L6 (Fig. 2c). The L6 CT neurons were divided into nine clusters (Fig. 2a), which exhibited a complex spatial pattern with distinctions in both cortical depth and medial–lateral directions (Fig. 2d). L6b cells, which formed the innermost layer of the cortex, were subdivided into three clusters intermixed in space (Fig. 2a, e).

## A gradient across the IT neurons

The IT neurons constitute the largest branch of neurons in the imaged region, which span nearly the entire cortical depth from L2/3 to L6.

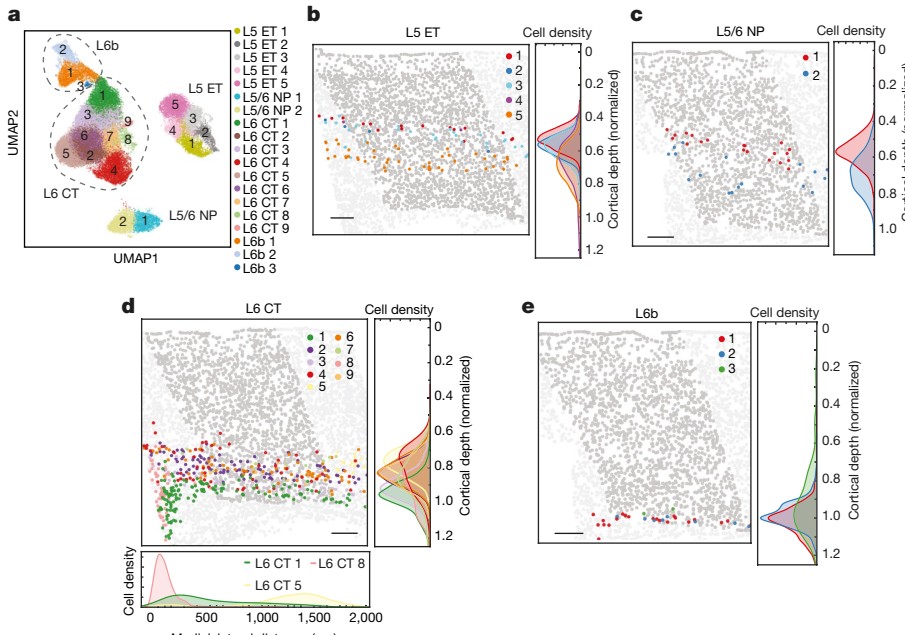

**Fig. 2 | Spatial organization of the L5 ET, L5/6 NP, L6 CT and L6b neurons.**
**a**, Uniform manifold approximation and projection (UMAP) of the L5 ET, L5/6 NP, L6 CT and L6b neurons coloured by the cluster identity. **b**, Spatial map of L5 ET cell clusters in a coronal slice (Bregma approximately +0.8). L5 ET cells are coloured, and other cells in the MOp are shown in dark grey to highlight the MOp region, whereas other cells outside the MOp are shown in light grey. Normalized cortical depth distributions of the cells of each L5 ET cluster are shown in the right panel and presented in the form of the Kernel density of the distribution histograms. **c**, Spatial map of L5/6 NP cell clusters in a coronal slice (Bregma approximately +0.4) and the normalized cortical depth distributions of these clusters, as in **b**. **d**, Spatial map of the L6 CT cell clusters in a coronal slice (Bregma approximately +0.3) and the normalized cortical depth distributions of these clusters, as in **b**. The medial–lateral distribution for clusters 1, 5 and 8 are also shown at the bottom. The medial–lateral distribution is calculated in the coronal slices in which cluster 8 is present. **e**, Spatial map of the L6b clusters in a coronal slice (Bregma approximately +0.4) and the normalized cortical depth distributions of these clusters, as in **b**. Scale bars, 200 μm (**b**–**e**).

Our MERFISH data classified the IT cells into 20 clusters, 19 of which belonged to the L2/3 IT, L4/5 IT, L5 IT and L6 IT subclasses and the remaining one formed a distinct cell type, L6 IT *Car3* (Fig. 3a, Extended Data Fig. 7a–c).

The L2/3, L4/5, L5 and L6 IT subclasses showed laminar organizations (Fig. 3b, c). These subclasses were each subdivided into several clusters, further parcellating each cortical layer but without discrete boundaries (Fig. 3b, c). Notably, MERFISH data identified seven IT clusters residing between the L2/3 and L5 IT neurons (Fig. 3c), expressing the L4 maker genes *Rspo1* and/or *Rorb* (Extended Data Fig. 2a) and corresponding to the L4/5 IT clusters identified by scRNA-seq and snRNA-seq[23] (Extended Data Fig. 7a). Among these clusters, L4/5 IT SSp 1 and 2 were located primarily in the neighbouring SSp region with a relatively minor presence in the MOp (Extended Data Fig. 7d), whereas L4/5 IT 1–5 showed substantial presence in the MOp (Fig. 3c). The MOp has been traditionally considered lacking a distinct L4 due to the absence of clear cytoarchitecture features[36]. Our results suggest the presence of L4 neurons in the MOp, consistent with a previous report of L4 neurons in the MOp based on their anatomical and connectivity properties[37].

We observed a largely gradual transition of the gene expression profiles among the IT clusters, except for L6 IT *Car3* (Fig. 3a, Extended Data Fig. 8a). Moreover, along the direction of the cortical depth, individual IT clusters partially overlapped in space with adjacent clusters, as evident from both the cell-type spatial maps of individual coronal slices (Fig. 3c) and the quantitative analysis of cortical depth distributions (Fig. 3b, Extended Data Fig. 8b–d). The lack of clear separation among the IT clusters led us to further evaluate whether the IT cells traverse a continuous spatial and molecular landscape. Quantification of the degree of intercluster connectivity[38] showed that the IT clusters formed an interconnected network with clusters exhibiting the highest connectivity (namely, highest similarity in gene expression) to those that

were spatially adjacent (Fig. 3d, Extended Data Fig. 9a). Identification of genes whose expression changed substantially with cortical depth revealed a largely gradual change of gene expression profiles of cells along the cortical depth axis, with steeper changes at the cortical depths that approximately separate cell subclasses (Fig. 3e, Extended Data Fig. 9b). Using pseudotime analysis[39] to order the IT cells on the basis of their expression profiles, we observed that the pseudotime of cells was highly correlated with their cortical depths, and individual cells formed a largely continuous cloud along the pseudotime and cortical depth axes, with a more appreciable separation in pseudotime between L2/3 and L4/5 IT clusters (Fig. 3f, Extended Data Fig. 9c, d). Together, these results suggest that the IT neurons adopt a gradient distribution across the cortical depth, with correlated gene expression profiles and cortical depths of individual cells.

## Projection pattern of IT neurons

We next sought to integrate MERFISH with retrograde tracing to simultaneously determine the expression profiles and spatial organization of cell types in the MOp and their projection targets. To this end, we injected retrograde tracers, cholera toxin subunit b (CTb) labelled with spectrally distinct dyes, into three cortical regions, the ipsilateral MOs, the SSp and the temporal association area (TEa), all of which receive direct inputs from the MOp[40,41]. TEa injections also spread into its adjacent ectorhinal (ECT) and perirhinal (PERI) areas, and these areas are often referenced together as a complex (TEa–ECT–PERI) in retrograde tracing studies[31,40]. Next, we identified neurons in the MOp that projected to these target regions by CTb imaging, followed by imaging the 258-gene MERFISH panel for cell-type identification (Fig. 4a). We performed this analysis for the approximate MOp upper limb domain (Bregma 0 to +1.0) and imaged coronal slices at approximately

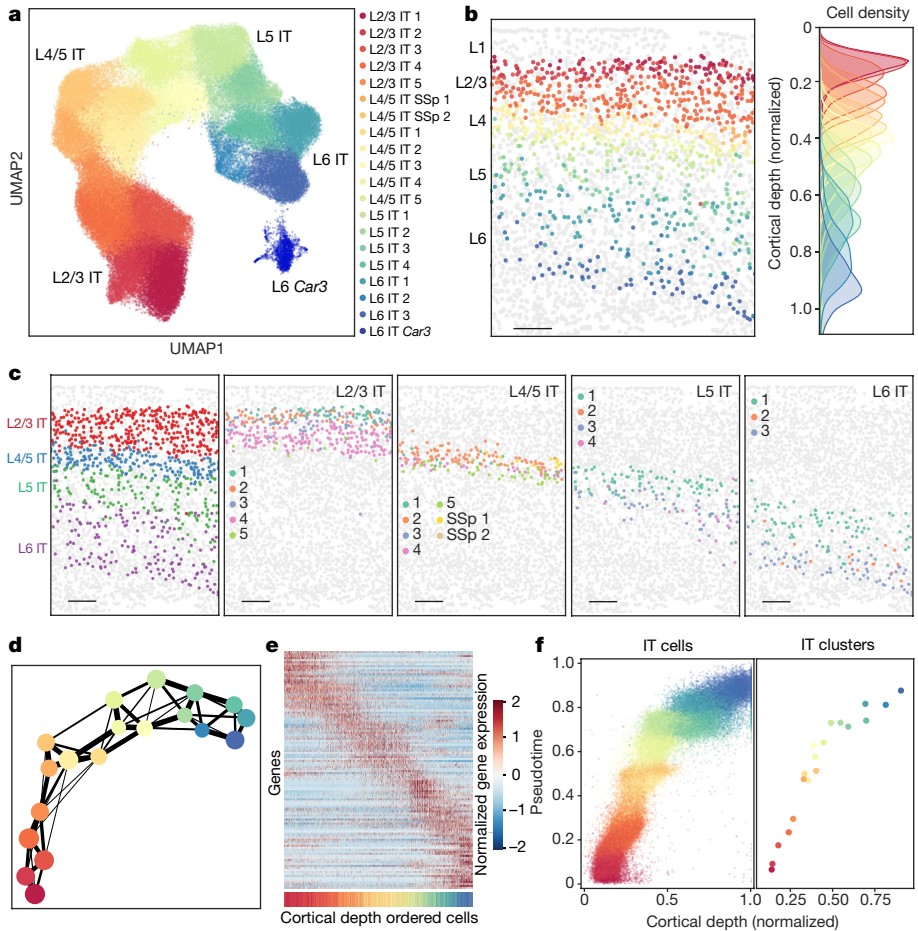

**Fig. 3 | Correlated gene expression and spatial gradients across IT neurons.** Analyses of IT neurons in the entire imaged region are shown here and the corresponding analyses for the MOp are shown in Extended Data Figs. 8, 9. **a**, UMAP of the IT clusters coloured by the cluster identity. **b**, Spatial map of IT neurons in a coronal slice (Bregma approximately +1.0). The IT neurons are coloured by their cluster identity as in **a**, and all other cells are in grey (left). Scale bar, 200 μm. The cortical depth distributions of individual IT clusters are also shown (right). **c**, Spatial maps of the L2/3, L4/5, L5 and L6 IT subclasses (left panel) and individual clusters in each subclass (four right panels) in the same coronal slice as in **b**. Scale bars, 200 μm. **d**, The degree of connectivity between clusters in a k-nearest neighbour graph for the IT clusters, with each cluster represented as a node coloured as in **a**, and the weighted edges between nodes representing their connectivity. Edges with weights below 0.1 are not shown. **e**, Normalized expression of differentially expressed genes of all IT neurons across cortical depth. Here, differentially expressed genes refer to genes of which the expression varied substantially with cortical depth (Methods). Individual IT neurons were sorted in the order of ascending cortical depth and the genes were sorted by the cortical depth at which they exhibit maximal expression. The coloured bar at the bottom indicates the cluster identity of the cell, coloured as in **a**. **f**, Scatter plot of the pseudotime versus normalized cortical depth for individual IT neurons (left) and individual IT clusters (right) in the L2/3, L4/5, L5 and L6 IT subclasses coloured by the cell clusters, as in **a**.

30-μm intervals along the anterior–posterior axis in two mice (approximately 190,000 cells total). We observed that approximately 90% of MOs-projecting, SSp-projecting and TEa–ECT–PERI-projecting neurons were IT and L6b neurons. Spatially, the MOs-projecting and SSp-projecting neurons were more broadly distributed along the cortical depth axis, whereas the TEa–ECT–PERI-projecting neurons showed a distinct multi-laminar distribution (Fig. 4b), consistent with previous observations[40].

We observed a complex projection pattern of MOp neurons. Neurons in the same cell clusters sent output to multiple targets (Fig. 4c) and, likewise, the same target region received inputs from multiple subclasses and clusters of MOp neurons (Fig. 4d). All three regions received inputs from a large number of individual clusters, each region from a quantitatively different composition of clusters (Fig. 4d, bottom panel).

Interestingly, some molecularly and spatially similar IT clusters showed distinct projection patterns. For example, almost all CTb-positive L6 IT 3 neurons projected to TEa–ECT–PERI but not to the MOs and the SSp, whereas the majority of the CTb-positive L6 IT 1 neurons projected to the MOs with very few to TEa–ECT–PERI (Fig. 4c).

Among the three L6 IT clusters, the MOs mostly received input from L6 IT 1, whereas TEa–ECT–PERI mostly received input from the L6 IT 3, despite the similar gene expression profiles and the substantially overlapping spatial distributions of these L6 IT clusters (Fig. 4e).

## Discussion

Here we used MERFISH to generate a molecularly defined and spatially resolved map of cell populations for the MOp and its adjacent areas in the mouse brain. The cell census defined by MERFISH, including 95 neuronal and non-neuronal populations, showed good correspondence to that defined by the single-cell sequencing in the companion study[23] and revealed distinct spatial distributions for most transcriptomically distinct cell populations. Our results showed laminar restrictions for different subclasses of neurons that are consistent with previous findings[12,13,28,29], but also revealed a previously unknown, high-resolution spatial map for individual neuronal clusters. We observed laminar organization not only for excitatory neurons but also for inhibitory neurons, with many inhibitory neuronal clusters preferentially located in one or two cortical layers. Moreover, many excitatory neuronal clusters

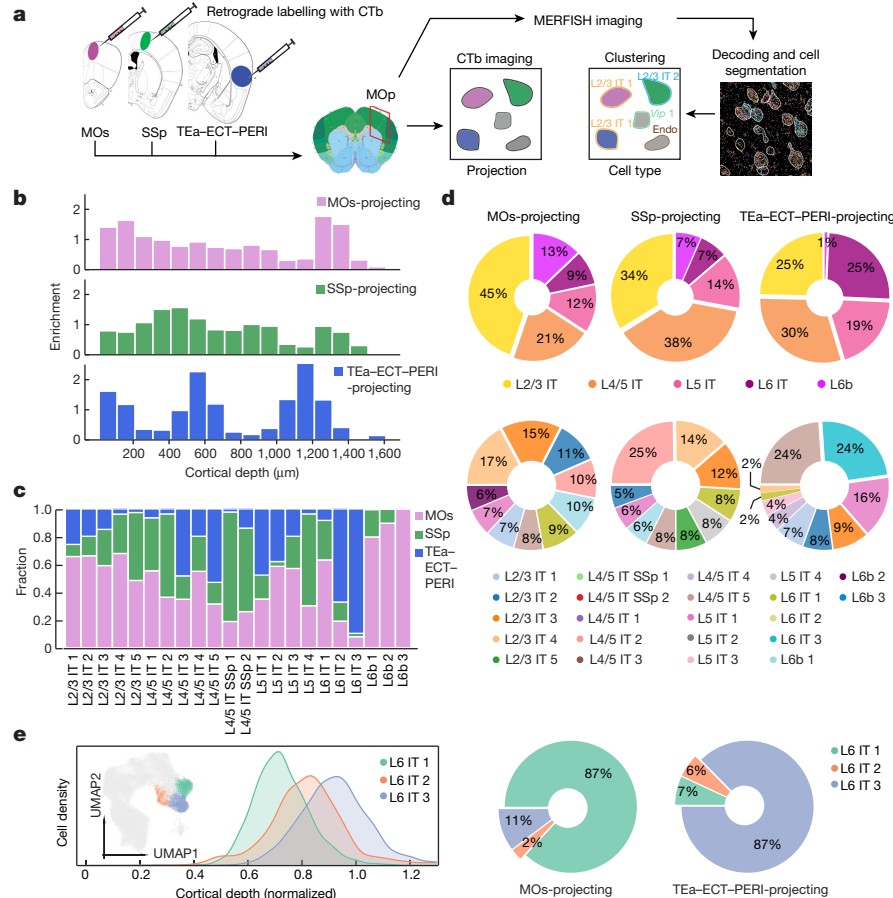

**Fig. 4 | Projection patterns of IT neurons determined by integration of MERFISH with retrograde labelling. a**, Workflow integrating retrograde labelling and MERFISH. CTb-AlexaFluor647, CTb-AlexaFluor555 and CTb-AlexaFluor488 were injected into the MOs, SSp and TEa–ECT–PERI regions, respectively. The coronal slices containing the MOp on the ipsilateral side of these targets were imaged for both the retrograde CTb labels and the MERFISH gene panel. The mouse brain CCF image shown on the left is from the Allen Brain Atlas (http://atlas.brain-map.org/; credit: Allen Institute). Endo, endothelial cell. **b**, Enrichment of MOs-projecting, SSp-projecting and TEa–ECT–PERI-projecting cells at different cortical depths. Enrichment is defined as the fraction of relevant CTb-positive cells divided by the fraction of all IT and L6b cells in the same bin. **c**, Fractions of MOs-projecting, SSP-projecting and

TEa–ECT–PERI-projecting cells in each cell cluster among all CTb-positive, single-projecting cells in the cluster. **d**, Pie charts showing the proportions of MOs-projecting (left), SSp-projecting (middle) and TEa–ECT–PERI-projecting (right) cells belonging to each cell subclass (top) and cluster (bottom; only top 10 clusters shown). The mean fractions are shown and the 95% confidence intervals are less than 0.7%. **e**, Projection specificity of the molecularly and spatially similar L6 IT clusters. The cortical depth distributions of the L6 IT 1–3 clusters and the UMAP (inset) are displayed, with L6 IT 1–3 neurons shown in colours and other IT neurons shown in grey (left). Pie charts showing the relative proportions of MOs-projecting and TEa–ECT–PERI-projecting L6 IT neurons that belong to each of the three clusters are also displayed (right). The mean fractions are shown and the 95% confidence intervals are less than 1%.

adopted narrow distributions along the cortical depth direction that revealed finer laminar structures within individual cortical layers.

We noticed that, although neurons tended to form discrete populations of cells with distinct expression profiles at the subclass level, clusters within individual subclasses often exhibited more gradual changes, adding evidence to the coexistence of discrete and continuous cell heterogeneity in the brain[13,42,43]. In particular, IT neurons, which constitute approximately 70% of all excitatory neurons in the MOp, formed a largely continuous gradient across the cortical depth. Continuous variations in gene expression have also been observed among IT neurons in the isocortex by a concurrent scRNA-seq study[44]. Here, with spatially resolved single-cell profiling, we observed correlated changes in the gene expression and cortical depth of IT neurons, revealing a molecular and spatial gradient of cells spanning nearly the entire cortical depth. It remains an open question whether other properties (for example, input or output connectivity) of the IT neurons could have more discrete layer specificity and, if so, whether these properties correlate with molecular signatures that are to be identified.

We further investigated how individual molecularly identified cell types correlate with their projection targets by integrating MERFISH with retrograde labelling. Our results showed that projections of MOp neurons to other cortical regions formed a complex multiple-to-multiple network: each cell cluster projects to multiple target regions (consistent with previous observations for visual cortex projections[45]), and each target region receives inputs from many clusters. We also observed distinct projection properties from some similar neuronal clusters with gradually varying expression profiles and overlapping spatial distributions. How such distinct projection properties arise from these similar clusters, whether it is due to a molecular signature not captured by transcriptomic profiling or has arisen from a developmental origin, remains an open question. Our proof-of-principle measurements probed only three target regions, but more target regions could be measured using this approach to construct a more comprehensive projection map for the cell types in the MOp. We envision that MERFISH may also be combined with trans-synaptic viral tracers to generate a high-resolution cell-type-to-cell-type connectivity map.

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

## Methods

### Animals

Adult C57BL/6J male mice aged 57–63 days were used in this study. Two mice were used for in situ cell-type identification and spatially mapping by MERFISH; and two animals were used for projection mapping by combining MERFISH with retrograde labelling. Mice were maintained on a 12-h light/12-h dark cycle (14:00 to 02:00 dark period), at a temperature of $22 \pm 1\,°C$, a humidity of 30–70%, with ad libitum access to food and water. Animal care and experiments were carried out in accordance with NIH guidelines and were approved by the Harvard University Institutional Animal Care and Use Committee (IACUC) and the University of South California Institutional Animal Care and Use Committee.

### Gene selection for MERFISH

To discriminate transcriptionally distinct cell populations with MERFISH, we designed a panel of 258 genes. Among the 258 genes, 62 were manually picked marker genes including established markers for inhibitory and excitatory neurons, as well as different non-neuronal cell markers for oligodendrocytes, oligodendrocyte precursor cells, astrocytes, microglia, perivascular macrophages, endothelial cells, pericytes, smooth muscle cells and vascular leptomeningeal cells (VLMCs). To further discriminate different neuronal cell types, we combined two approaches to select genes based on clustering results from scRNA-seq and snRNA-seq data. In the first approach, we selected a panel of genes with the highest mutual information as previously reported[22]. Briefly, we used mutual information to determine the relative amount of information each gene carries in defining the clusters identified by scRNA-seq and snRNA-seq. We used the scRNA-seq 10x v2 A dataset generated by a companion study[23] and determined highly variable genes using the Scanpy[46] package. We binarized the expression profiles using a gene counts cut-off of zero to simplify the calculation of the mutual information. We selected the top 50 genes with the highest mutual information for excitatory and inhibitory neuronal clusters, respectively, and due to overlap between the two groups, this approach generated a total of 91 top mutual information genes. In the second approach, we selected a panel of 168 genes based on differentially expressed (DE) gene analysis using the scRNA-seq data (scRNA-seq 10x v2 and scRNA-seq SMART data) from the companion study[23]. We first found DE genes for each neuronal cluster pair (consisting of a foreground cluster and a background cluster) in both directions. The criteria to define DE genes were: the genes have a twofold or more change in expression between the foreground and background clusters and $P < 0.05$; they are expressed in at least 40% of cells in the foreground cluster, with more than threefold enrichment, in terms of the fraction of cells expressing the gene, relative to the background cluster. $P$ values were calculated using the analysis of variance (ANOVA) test in limma[47] on log-transformed data. The top 50 genes that passed all of the tests and ranked by $P$ values in each direction for every cluster pair were pooled together as candidates for scoring for the final marker set. To determine the final marker list, which we required to include at least two genes in each direction for all pairs of clusters, we used a greedy algorithm to find the minimal number of genes that satisfied the requirement. Starting from a manually picked marker gene list as described above, the algorithm checks which pairs already have a sufficient number of DE genes, and works on the remaining pairs of clusters until each pair of clusters has at least two DE genes included in each direction. This approach generated a total of 168 genes.

We note that the mutual information genes tend to be genes that are differentially expressed between groups of cell clusters, whereas the DE genes are differentially expressed between individual pairs of clusters. These two sets have complementary power and, when combined, can give better cluster identification results in our experience. We thus combined the marker gene lists generated by these three different approaches, which partially overlap with each other, resulting in a panel of 258 genes in total. We then screened this gene list to identify genes that are relatively short or have relatively high expression level, which were potentially challenging for highly multiplexed FISH imaging experiments, as previously described[21]. We found 16 genes that can accommodate fewer than 48 hybridization probes with target sequences that are 30-nucleotides (nt) long, or are expressed at an average of 200 or greater counts per cell in any cell cluster as determined from the scRNA-seq SMART data[23]. These 16 genes were imaged in a set of eight sequential, two-colour FISH imaging rounds, following the MERFISH run that imaged the remaining 242 genes.

### Design and construction of the MERFISH encoding probes

MERFISH encoding probes for the 242 genes were designed as previously described[21]. We first assigned to each of the 242 genes a unique binary barcode drawn from a 22-bit, Hamming-Distance-4, Hamming-Weight-4 encoding scheme. We included 10 extra barcodes as 'blank' barcodes, which were not assigned to any genes, to provide a measure of the false-positive rate in MERFISH as previously described[21].

We identified all possible 30-mer targeting regions within each desired gene transcript as previously described[48]. Each MERFISH encoding probe contains a 30-mer targeting region that is complementary to the RNA of interest, as well as two 20-mer readout sequences that define the specific barcode assigned to each gene. From the set of all possible 30-mer targeting sequences for each gene, we selected 92 30-mer targeting sequences at random. For the transcripts that were not long enough and had fewer than 92 targeting sequences, we allowed these 30-mers to overlap by as much as 20 nt to increase the number of possible encoding probes – because a given transcript is typically bound by less than one-third of the 92 encoding probes[49], the encoding probes with overlapping targeting regions do not substantially interfere with each other but partially compensate for reduced binding due to local inaccessible regions on the target RNA or loss of probe during synthesis. We then assigned two readout sequences to each of the encoding probes associated with each gene. For the 22-bit encoding scheme, a total of 22 readout sequences were used, each corresponding to 1 bit, and the collection of encoding probes for each gene contained 4 of the 22 readout sequences that corresponded to the 4 bits that reads '1' in the barcode assigned to that gene.

Encoding probes for the 16 genes imaged in sequential two-colour FISH rounds were produced in the same manner, except that 48 targeting sequences were selected, and one single unique readout sequence was included in each set of the 48 targeting sequences. The readout sequences used here were different from the 22 readout sequences used for the genes detected in the MERFISH run.

In addition, we concatenated to each encoding probe sequence two PCR primers, the first comprising the T7 promoter, and the second being a random 20-mer designed to have no region of homology greater than 15 nt with any of the encoding probe target sequences designed above, as we previously described[48].

With the above-described template encoding probe sequences, we constructed the MERFISH probe set as previously described[21]. The template DNA were synthesized as a complex oligo pool (Twist Biosciences). This pool contained both the encoding probes to the 242 genes measured in the MERFISH run and the 16 genes measured in the sequential two-colour FISH rounds, but different primer sequences for the two sets, which allowed us to amplify these two probe sets separately via PCR followed by the same synthesis and purification procedures. The two probe sets were then mixed during tissue staining.

### Design and construction of MERFISH readout probes

For the 258-gene panel used in this study, 38 readout probes were designed, each complementary to one of the 38 readout sequences. Twenty-two of the 38 readout probes correspond to the 22 bits in the barcodes used for MERFISH imaging, and the remaining 16 readout

probes each corresponds to one gene that was imaged in the sequential two-colour FISH rounds. Each readout probe was conjugated to one of the two dye molecules (Alexa750 or Cy5) via a disulfide linkage, as previously described[48]. These readout probes were synthesized and purified by Bio-Synthesis, Inc., resuspended immediately in Tris-EDTA (TE) buffer, pH 8 (Thermo Fisher), to a concentration of 100 µM and stored at −20 °C.

## Tissue preparation for MERFISH

Mice aged 57–63 days were euthanized with $CO_2$; their brains were quickly harvested and cut into hemispheres and each hemisphere was frozen immediately on dry ice in optimal cutting temperature compound (Tissue-Tek O.C.T.; 25608-930, VWR), and stored at −80 °C until cutting. Frozen brain hemispheres were sectioned at −18 °C on a cryostat (Leica CM3050 S). Slices were removed and discarded until the MOp region was reached. A continuous set of 300, 10-µm-thick slices were cut from anterior to posterior, and approximately every tenth slice was placed onto coverslips for imaging. Each coverslip contained 4–6 slices. The coverslips were prepared as previously described[21,50].

Tissue slices were fixed by treating with 4% PFA in 1× PBS for 15 min and were washed three times with 1× PBS and stored in 70% ethanol at 4 °C for at least 18 h to permeabilize cell membranes. The tissue slices from the same mouse were cut at the same time and distributed to six coverslips, which were store in 70% ethanol at 4 °C for no longer than 2 weeks until all the coverslips were imaged. We observed no degradation in sample quality over this time period.

The tissue slices were stained with the MERFISH probe set as previously described[21]. Briefly, the samples were removed from the 70% ethanol and washed with 2× saline sodium citrate (SSC) three times. Then, we equilibrated the samples with encoding-probe wash buffer (30% formamide in 2× SSC) for 5 min at room temperature. The wash buffer was then aspirated from a coverslip, and the coverslip was inverted onto a 50-µl droplet of encoding-probe mixture on a parafilm-coated Petri dish. The encoding-probe mixture comprised approximately 1 nM of each encoding probe for the MERFISH run, approximately 5 nM of each encoding probe for the sequential two-colour FISH rounds and 1 µM of a polyA-anchor probe (IDT) in 2× SSC with 30% v/v formamide, 0.1% wt/v yeast tRNA (15401-011, Life Technologies) and 10% v/v dextran sulfate (D8906, Sigma,). We then incubated the sample at 37 °C for 36–48 h. The polyA-anchor probe containing a mixture of DNA and LNA nucleotides (/5Acryd/TTGAGTGGATGGAGTGTAATT+TT+TT+TT+TT+TT+TT+TT+ TT+TT+T, where T+ is locked nucleic acid, and /5Acryd/ is 5′ acrydite modification) hybridized to the polyA sequence on the polyadenylated mRNAs and allowed these RNAs to be anchored to a polyacrylamide gel as described below. After hybridization, the samples were washed in encoding-probe wash buffer for 30 min at 47 °C for a total of two times to remove excess encoding probes and polyA-anchor probes. All tissue samples were cleared to remove fluorescence background as we previously described[21,50]. Briefly, the samples were embedded in a thin polyacrylamide gel and were then treated with a digestion buffer of 2% v/v sodium dodecyl sulfate (SDS; AM9823, Thermo Fisher), 0.5% v/v Triton X-100 (X100, Sigma) and 1% v/v proteinase K (P8107S, New England Biolabs) in 2× SSC for 36–48 h at 37 °C. After digestion, the coverslips were washed in 2× SSC for 30 min for a total of four washes and then stored at 4 °C in 2× SSC supplemented with 1:100 murine RNase inhibitor (M0314S, New England Biolabs) before imaging.

## MERFISH imaging

We used a home-built imaging platform in this study as previously described[20]. To prepare the sample for imaging, we first stained it with a readout hybridization mixture containing the readout probes associated with the first round of imaging in the MERFISH run, as well as a probe complementary to the polyA-anchor probe and conjugated via a disulfide bond to the dye Alexa488 at a concentration of 3 nM. The readout hybridization mixture comprised the readout-probe wash buffer comprised 2× SSC, 10% v/v ethylene carbonate (E26258, Sigma) and 0.1% v/v Triton X-100, supplemented with 3 nM each of the appropriate readout probes. The sample was incubated in this mixture for 15 min at room temperature, and then washed in the readout-probe wash buffer supplemented with 1 µg/ml DAPI for 10 min to stain nuclei within the sample. The sample was then washed briefly in 2× SSC and imaged. Briefly, the sample was loaded into a commercial flow chamber (FCS2, Bioptechs) with a 0.75-mm-thick flow gasket (DIE# F18524; 1907-100, Bioptechs). Imaging buffer comprising 5 mM 3,4-dihydroxybenzoic acid (P5630, Sigma), 2 mM trolox (238813, Sigma), 50 µM trolox quinone, 1:500 recombinant protocatechuate 3,4-dioxygenase (rPCO; OYC Americas), 1:500 murine RNase inhibitor and 5 mM NaOH (to adjust pH to 7.0) in 2× SSC was introduced into the chamber and the sample was imaged with a low-magnification objective (CFI Plan Apo Lambda ×10, Nikon) with 405-nm illumination to produce a low-resolution mosaic of all slices in the DAPI channel. We then used this mosaic image to locate the MOp region in each slice and generated a grid of field-of-view (FOV) positions to cover the MOp region and adjacent areas to be imaged. We then switched to a high-magnification, high-numerical aperture objective (CFI Plan Apo Lambda ×60, Nikon) and imaged each of the FOV positions generated above. In the first round of imaging, we collected images in the 750-nm, 650-nm, 560-nm, 488-nm and 405-nm channels to image the first two readout probes (conjugated to Alexa750 and Cy5, respectively), the orange fiducial beads, the total polyA mRNA stained by the polyA-anchor probe (Alexa488) and the nucleus stained by DAPI (405-nm channel). The latter two channels were used for cell segmentation as described below. We took a single image for the fiducial beads on the surface of the coverslip using the 560-nm illumination channel for each imaging round as a spatial reference to correct for slight misalignments in the stage position over the imaging rounds. To image the entire volume of each 10-µm-thick slice, we collected seven 1.5-µm-thick z-stacks for the other four channels (two readout probes, polyA probe and DAPI) in each FOV.

After the first round of imaging, the dyes were removed by flowing 2.5 ml of cleavage buffer comprising 2× SSC and 50 mM of Tris (2-carboxyethyl) phosphine (TCEP; 646547, Sigma) with 15-min incubation in the flow chamber, to cleave the disulfide bond linking the dyes to the readout probes. The sample was then washed by flowing 1.5 ml 2× SSC.

To perform subsequent rounds of imaging, we flowed 3.5 ml of the readout probe mixture containing the appropriate readout probes across the chamber and incubated the sample in this mixture for a total of 15 min for each round. The sample was then washed by 1.5 ml of readout-probe wash buffer and then 1.5 ml of imaging buffer was introduced into the chamber. For each round, we took images for all FOV locations in the 750-nm, 650-nm and 560-nm channels for the two readout probes and fiducial beads. Two readout probes were imaged in each round, one labelled with Alexa750 and the other with Cy5, and a readout-probe mixture containing 3 nM of appropriate readout probes was used for each round. We repeated the hybridization, wash, imaging and cleavage for all rounds to complete the 22-bit MERFISH imaging and the eight rounds of sequential two-colour FISH. All buffers and readout-probe mixtures were loaded with a home-built, automated fluidics system composed of three 12-port valves (EZ1213-820-4, IDEX) and a peristaltic pump (MP3, Gilson), configured as previously described[5]. The total MERFISH imaging time was approximately 24–36 h for each experiment, which contained 4–6 coronal slices.

## MERFISH image analysis and cell segmentation

All MERFISH image analysis was performed using MERlin[51], a Python-based MERFISH analysis pipeline, using algorithms similar to what we have previously described[20,21]. First, we aligned the images taken during each imaging round based on the fiducial bead images, accounting for X–Y drift in the stage position relative to the first round of imaging. For the MERFISH images, we then high-pass filtered the

image stacks for each FOV to remove background, deconvolved them using 20 rounds of Lucy–Richardson deconvolution to tighten RNA spots, and low-pass filtered them to account for small movements in the apparent centroid of RNAs between imaging rounds. Individual RNA molecules were identified by our previously published pixel-based decoding algorithm[48]. After assigning barcodes to each pixel independently, we aggregated adjacent pixels that were assigned with the same barcodes into putative RNA molecules, and then filtered the list of putative RNA molecules to enrich for correctly identified transcripts as previously described[20] for an overall barcode misidentification rate at 5%. We further removed putative RNAs that contained only a single pixel as they are prone to be background of spurious barcodes generated by random fluorescent fluctuations and had a substantially higher misidentification rate than those that contained 2 or more pixels.

We segmented cell boundaries in each FOV using a seeded watershed approach as previously described[21]. The DAPI images were used as seeds and the polyA signals were used to identify segmentation boundaries. Finally, we assigned individual RNA molecules identified in the MERFISH run to individual cells based on whether they fell within the segmented boundaries of the cells. For the sequential two-colour FISH rounds, we quantified the signal from these imaging rounds by summing the fluorescence intensity of all pixels that fell within the segmentation boundaries of the cells associated with the central $z$-plane and normalized the signal by the areas of the cells in this $z$-plane. Then, the normalized signals of the 16 genes from the sequential two-colour FISH rounds were merged with the RNA count matrix from the 242 genes measured in the MERFISH run and used for cell clustering analysis.

## Cell clustering analysis of MERFISH data

With the cell-by-gene matrix obtained as described above (each row representing a cell and each column representing a gene, and each element representing the expression level a specific gene in a specific cell), we preprocessed the matrix by the following steps. (1) The segmentation approach that we used generated a small fraction of putative 'cells' with very small total volumes due to spurious segmentation artefacts, as well as some cells that overlapped in the 3D and were not properly separated. We hence removed the segmented 'cells' that had a volume that was either less than 100 μm³ or larger than three times of the median volume of all cells. (2) A fraction of cells did not have the whole soma included in a 10-μm-thick tissue slice and was thus not imaged completely. To remove the differences in RNA counts due to the incompleteness of the imaged soma volume, we normalized the RNA counts per cell by the imaged volume of each cell. (3) We observed a modest batch effect between MERFISH experiments accounting for approximately 30% variation of the mean total number of RNAs per cell. We normalized the mean total RNA counts per cell to a same mean value (250 in this case) for each experiment to remove the influence of these batch effects. (4) Since the 16 genes that were imaged in the sequential FISH rounds contained many specific marker genes that should not co-express in individual cell types, and no cells should express a majority of these 16 genes, we considered the segmented 'cells' that had a normalized fluorescence signal that was higher than the 90% quantile in 12 out of the total 16 sequential FISH channels as caused by spurious fluorescence background and removed them. (5) Since the fluorescence background in the 650-nm and 750-nm channels was different, we subtracted the background for each cell by taking the minimum of the signal for each cell across all sequential FISH rounds as the background, for 650-nm and 750-nm channels separately. (6) We removed the cells that had total RNA counts lower than 2% quantile or higher than 98% quantile. (7) We removed potential doublets using Scrublet[52]. Briefly, principal component analysis (PCA) was used to train a $k$-nearest neighbour (kNN) classifier to predict a doublet score for each cell. Since we recorded the DAPI-stained nucleus image of each cell, we were able to visually inspect a random subset of potential doublets picked by Scrublet and fine-tuned the doublet score

threshold to remove connected cells more accurately. Finally, the cells with a doublet score higher than 0.18 were removed, which accounted for approximately 12% of the total cell number. (8) We also found that 4 out of the 16 genes imaged in the sequential FISH rounds − Cd52, Rprml, Mup5 and Igfbp6 − were not stained well in all experiments and failed to yield high-quality signals. These four genes were removed for subsequent analysis.

After the above preprocessing steps, we normalized the total RNA counts for each cell to the median total RNA counts of all cells and log-transformed the cell-by-gene matrix. We then normalized their expression profiles by computing the $z$-score for each gene. We performed dimensionality reduction of the matrix using PCA, and used the first 35 principal components. To determine the number of principal components to keep, we randomly shuffled the values in each column of the cell-by-gene matrix and calculated the eigenvalue of the first principal component for the randomly shuffled matrix. The random shuffling was repeated 20 times and the mean eigenvalue of the first principal component across 20 iterations was obtained, and we kept all of the principal components that had an eigenvalue greater than this mean value. We then performed graph-based Louvain community detection[53] in the 35 principal components space using Scanpy[46] for a range of nearest neighbourhood size $k$ values with a bootstrap analysis to both identify stable clusters and select the optimal $k$ value ($k = 10$) as previously described[21]. We further identified six small clusters that expressed mixtures of markers for multiple distinct cell classes, for example, Slc17a7, which marks excitatory neurons, and Sox10, which marks the oligodendrocytes, and that did not correspond to any of the major subclasses defined by the scRNA-seq and snRNA-seq data[23] (based on classifier analysis, which is described below), as potential doublets, which were excluded from subsequent analysis.

From the first round of clustering, we identified 16 excitatory neuronal clusters, 8 inhibitory neuronal clusters and 14 other clusters. To further refine our detection of transcriptionally distinct populations, we separated all of the cells into five groups: IT-projecting neurons (marked by the excitatory neuronal marker Slc17a7 and the pan-IT marker Slc30a3), non-IT neurons (marked by the excitatory neuronal marker Slc17a7 but not Slc30a3), caudal ganglionic eminence (CGE)-derived inhibitory neurons (marked by Gad1, Gad2 and Lamp5/Sncg/Vip), medial ganglionic eminence (MGE)-derived inhibitory neurons (marked by Gad1, Gad2 and Sst/Pvalb) and non-neuronal cells. We then repeated the procedure of dimensionality reduction and clustering, as described above, for these five cell groups separately. In addition, we sampled a range of resolution parameter $r$ ($r = 1, 2, 3$), a parameter value defined in Scanpy[46] that controls the coarseness of the clustering, to search for optimal granularity that represents the diversity of the transcriptomic profiles. We kept $k = 40$ and $r = 2$ for IT and non-IT excitatory neurons, $k = 15$ and $r = 2$ for CGE-derived and MGE-derived inhibitory neurons, and $k = 20$, $r = 1$ for the non-neuronal cells.

After the second round of clustering, we further removed a small fraction of cells as potential doublets as described above. We also found four unique clusters that did not correspond to any subclass in the MOp region defined by the scRNA-seq and snRNA-seq data[23] (using the classifier approach described below). We located the cells that belonged to these clusters and found that two clusters were in the striatum, and the other two clusters were probably ependymal cells located in the lateral ventricle. We removed these clusters from subsequent analysis.

After the clustering was done, the cell clusters were first each assigned into a subclass based on their marker gene expression as described in the main text. The IT neurons were further divided into L2/3, L4/5, L5 and L6 subclasses based on the expression of layer-specific makers (Cux2, Otof, Rorb, Rspo1, Sulf2, Fezf2 and Osr1). Since these markers showed gradual changes between individual IT clusters, the subclass identification at the border of layers can be ambiguous, in which case, we identify the parent subclass for the cluster by judging both its marker gene expression and its strongest corresponding cluster

in the scRNA-seq and snRNA-seq data. For example, L4/5 IT 5 expressed both *Rorb* and *Fezf2*, and corresponded to the L4/5 IT 2 cluster in the scRNA-seq and snRNA-seq data, and was thus classified as a L4/5 IT cluster. After the subclass identity was assigned, within each subclass, a numerical index was added following the subclass name to form the cluster name (for example, L5 IT 1, astrocyte 2, and so on).

For presentation, UMAP[54] was used to embed the cells in two dimensions using the same principal components that were used for clustering.

### Correspondence between clusters identified by MERFISH and single-cell sequencing-based measurements

Correspondence between cell clusters identified by MERFISH and by scRNA-seq and snRNA-seq in Extended Data Fig. 2c, d was assessed by running a neural-net classifier[55], which was trained on the *z*-scored single-cell expression profiles measured by MERFISH. The snRNA-seq 10x v3 B data in the companion paper[23] were used for comparison because it is the largest dataset among the seven scRNA-seq and snRNA-seq datasets included in this companion study and contained the largest number of non-neuronal cells, while all of the other six datasets were collected by fluorescence-activated cell sorting (FACS) to enrich for neurons. The snRNA-seq 10x v3 B data were *z*-scored, and then the subset of genes measured in the MERFISH data was used together with the trained model to predict a MERFISH cluster label for each cell in the snRNA-seq dataset. From this, each snRNA-seq cell had both a predicted MERFISH cluster label and a cluster label determined from the consensus clustering results for the seven scRNA-seq and snRNA-seq datasets[23]. Cells were grouped based on their consensus scRNA-seq and snRNA-seq cluster identity, and then the fraction of cells from a given consensus scRNA-seq and snRNA-seq cluster that were predicted to have each MERFISH cluster label was then determined (Extended Data Fig. 2d). The same classifier approach was also used to produce Extended Data Fig. 2c, but in this case, the subclass labels defined by MERFISH and by the seven scRNA-seq and snRNA-seq datasets for each cell was used instead of cluster labels. Likewise, the same classifier approach was used to produce Extended Data Figs. 6a, b, 7a, b, but in Extended Data Figs. 6b, 7b, the cluster labels defined by the integrated analysis of the seven scRNA-seq and snRNA-seq datasets, a snATAC-seq dataset and a snmC-seq dataset were used instead of the cluster labels derived from the scRNA-seq and snRNA-seq datasets alone.

### Registration to the Allen Reference Atlas and the common coordinate framework

For each coronal section that we performed, high-resolution MER-FISH/DAPI/polyA imaging of the MOp and adjacent areas, we also performed lower-resolution DAPI imaging of the entire hemisphere. The low-resolution DAPI image of each hemisphere coronal section was manually paired with the closest matching coronal section of the Allen Reference Atlas (ARA)[27] based on cytoarchitectural features. Once paired, landmark cytoarchitectural features were used to calculate a deformable or affine transformation from our DAPI image to the Nissl template of the matching ARA coronal section. Segmented cell boundaries from high-resolution MERFISH imaging were then aligned to the corresponding low-resolution DAPI image by aligning the high-resolution and low-resolution DAPI images. The overall transformation from both steps then allowed registration of the MERFISH images to the ARA. Out of the 64 coronal slices imaged, 61 slices were registered to the ARA, whereas the remaining three slices did not have a sufficient number of landmarks to be registered.

To define the boundaries of the MOp in the MERFISH images, each ARA template was further scaled and aligned via translation and rotation to the corresponding 2D coronal image in the Allen common coordinate framework (CCF) v3[26], which in turn allowed the MERFISH images to be registered to the Allen CCF v3.

To estimate the errors in image registration, we determined for each slice the average displacement between the cells on the cortical surface and the top surface of the cortex in the CCF image, as well as the average displacement between the L6b cells and the bottom surface of the cortex in the CCF image, and calculated these displacements as a percentage of the cortical thickness in that slice. For the 61 registered slices, the alignment error was on average 2.5% when calculated using the mean of the absolute values of the top and bottom surface displacements. To further reduce the effect of the alignment error in delineating cells within the MOp, we removed the slices that had an alignment error that was approximately 7% or greater for either the top or bottom surface, or approximately 5% or greater for their mean. In total, eight slices were removed from subsequent analyses that involved MOp delineation, and the remaining slices on average had an alignment error of 2.0% when calculated using the mean of the absolute values of the top and bottom surface displacements.

Registration of the MERFISH images to the Allen CCF v3 allowed us to place the imaged and profiled cells in the CCF, delineating cells in different brain regions. This version of CCF was chosen by the BICCN consortium for multiple modalities of measurements of the MOp to provide consistency among these measurements. While the brain areal boundaries may not be perfectly determined in the CCF v3 and efforts in the community will continue to improve the accuracy of these boundaries, the MERFISH results reported here will continue to serve as a resource as these areal boundaries are improved over time.

### Soma depth determination

From the MERFISH images, we segmented the cells and determined the centroid coordinates of all the cells. For each cell, the soma depth was determined as the shortest distance of its centroid position to the cortical surface line, which is marked by the very thin layer of VLMCs. Hence, the soma depths of individual cells were determined along the direction perpendicular to the cortical surface line in each coronal slice. To compensate the variation in cortical thickness from slice to slice, we measured the cortical thickness in each coronal slice, which was defined as the median soma depths of the L6b cells in the slice, and the soma depth of each cell was normalized by the cortical thickness of the slice. Cortical depth distribution analyses were performed for the region between Bregma −0.8 and +1.7 because MERFISH images of slices at Bregma +1.8 or greater did not show L6b cells forming a thin layer, which made normalization of the soma depth by the cortical thickness challenging for these anterior-most slices (Bregma between +1.8 and +2.5).

### Layer boundary assessment

The layer boundaries along the normalized cortical depth axis were determined as follows: (1) the cortical surface was defined by the positions of surface VLMCs; (2) we calculated the normalized median cortical depth of all cell clusters and used the median depth of the most superficial L2/3 IT cluster, L2/3 IT 1, as the upper boundary of L2/3; (3) the median depth of the most superficial L4/5 IT cluster, L4/5 IT 1, was used as the upper boundary of L4; (4) the median depth of the most superficial cluster among the L6 IT and CT clusters, L6 IT 1, was used as the upper boundary of L6; (5) the median depth of the L6b cells was set to 1 (as the soma depth of all cells were normalized by the median soma depths of the L6b cells) and the upper and lower boundaries of L6b were determined by the width of the L6b cell distribution; and (6) we also used the median depth of the most superficial cluster among the clusters residing in L5 to mark the upper boundary of L5 (that is, the boundary between L4 and L5); however, this boundary has some uncertainty because some of the L4/5 IT clusters may belong to L5, as we discuss below.

To examine which of the L4/5 IT clusters might also belong to L5, we examined the spatial overlap of the IT clusters with the L4 marker gene *Rspo1* and the L5 marker gene *Fezf2*. To this end, we first determined the

spatial profile of each of the two marker genes by binning all imaged cells into 100 equal-sized bins based on the normalized cortical depth and determining the mean expression level per cell for each bin. For each IT cluster, its spatial overlap with these marker genes was then determined as the fraction of the cells in the cluster that fell within the cortical depth range where the binned median expression of the marker gene was above half maximum. We observed that the spatial overlap with the L4 marker *Rspo1* took a rather precipitous fall at the L4/5 IT 5 cluster, with the overlap between L4/5 IT 5 and *Rspo1* being substantially lower than those between L4/5 IT 1–4 clusters and *Rspo1*. In addition, the spatial overlap of the L4/5 IT 5 cluster with the L5 marker *Fezf2* was substantially higher than those of the L4/5 IT 1–4 clusters and comparable to those of several L5 IT clusters (Extended Data Fig. 10a, b). Hence, the L4/5 IT 5 cluster probably resided in (or partially resided in) L5, and we thus considered the region between the median cortical depth of L4/5 IT 5 and the median cortical depth of L5 IT 1 as the uncertainty region for the upper boundary of L5 as shown by the grey area in Fig. 1c–e and Extended Data Fig. 3.

The L5 can be divided into a superficial L5a sublayer devoid of ET cells and a deeper L5b sublayer occupied by ET cells[56]. We also examined the spatial overlap between L4/5 IT and L5 IT clusters with the L5 ET neurons to assess which clusters may belong to L5a. The spatial overlap between a cell cluster and the L5 ET cells was defined as the overlapping area of the cell density distributions of the cell cluster and the L5 ET cell subclass. We observed that the L4/5 IT 5 cluster showed minimal spatial overlap with L5 ET (Extended Data Fig. 10c, d) and hence may reside in L5a. L5 IT 1 partially overlaps with L5 ET, but the spatial overlap of L5 IT 1 with L5 ET cells was substantially lower than those of the other L5 IT clusters (Extended Data Fig. 10c), suggesting that the L5 IT 1 cluster may partially reside in L5a.

## Connectivity and pseudotime analyses of IT neurons

To visualize the degree of similarity (connectivity) in the gene expression profiles of the IT clusters, we employed a recently developed graph abstraction technique called PAGA[38] to gain a quantitative understanding of how extensively different IT clusters occupied overlapping gene expression space. To this end, we first took the 19 IT clusters in the L2/3 IT, L4/5 IT, L5 IT and L6 IT subclasses and normalized their expression profiles by computing the $z$-score for each gene. Cells from the L6 IT Car3 were not included in this analysis as it formed a cluster that was well-separated in gene expression from the other IT cell clusters. PCA was used to reduce dimensionality of the normalized expression data to the first 19 principal components. In selecting the number of principal components to include, we performed the same random shuffling procedure used when setting a PC threshold for cell clustering analysis as described in the 'Cell clustering analysis of MERFISH data' section. We then constructed a kNN graph based on the principal components, identifying the 12 nearest neighbours of each cell. Using the kNN graph and the cluster label of each cell, we used Scanpy[46] to calculate the frequency that edges from cells with a given cluster label were connected to cells from a different cluster label and then normalized this frequency to that expected by chance. The resulting values represent the connectivity between the clusters in the kNN graph, and are visualized in a graph where each cluster is a node and the edges between nodes indicate the connectivity between those clusters.

Next, we constructed an ordering of the IT cells based on their expression profiles, yielding a 'pseudotime' value for each cell. This calculation is most often performed to order cells within a dynamic system, in which case the ordering reflects the 'time' relative to some reference cell. Our pseudotime calculation performed on the IT cells is not intended to represent the trajectory from L2/3 to L6 as part of a dynamic process, but rather to obtain an expression-derived measure of where along the trajectory each cell falls. To calculate the pseudotime of the IT cells, we used Scanpy to construct a diffusion map based on the above-described kNN graph, assigned a neuron from the L2/3 IT 1 cluster as the root cell

of the trajectory, and then computed the diffusion-based pseudotime[57]. The resulting value assigned to each cell reflects how far from the root cell its expression profile places it, and since each cell falls along a single trajectory with the L2/3 IT root cell at one end, this value orders the cells relative to one another along this path.

To identify genes that vary as a function of the cortical depths of the IT cells, the expression profiles of the IT cells were normalized by computing the $z$-score for each gene. The IT cells were split evenly into 50 bins based on their normalized cortical depths, and the mean normalized expression was calculated for each gene across all the bins. Any gene for which the difference in mean normalized expression between any two bins exceeded 0.5 was selected as a gene differentially expressed across cortical depth. To plot these genes in a heatmap, the genes were ordered according to the normalized cortical depths at which they exhibit their maximum expression and the cells were ordered based on their normalized cortical depths. To determine the cortical depth at which each gene exhibits its maximum expression, a rolling average was calculated across the 50 bins, using a window size of 10 bins, and the window at which the maximum expression value occurred was determined.

## Stereotaxic injection of retrograde tracers

To retrogradely label MOs-projecting, SSp-projecting and TEa–ECT–PERI-projecting MOp neurons, each region was injected in the same mouse in the right hemisphere with 100 nl of fluorescently conjugated CTb (CTb-AlexaFluor488, CTb-AlexaFluor555 or CTb-AlexaFluor647, respectively; 0.5%; C22841, C22843, and C34778, Thermo Fisher) using the following coordinates relative to Bregma: MOs (anterior–posterior (AP) +2.4 mm, medial–lateral (ML) +1.0 mm, dorsal–ventral (DV) +0.4 mm below the cortical surface), SSp (AP −0.5 mm, ML +2.4 mm, DV +0.5 mm below the cortical surface) and TEa–ECT–PERI (AP −1.7 mm, ML +4.5 mm, DV +2.5 mm below the cortical surface). Injection procedures were performed in adult male C57BL/6J mice (Jackson Laboratories) aged 2–4 months. Briefly, mice were anaesthetized initially in an induction chamber containing 5% isoflurane mixed with oxygen and then transferred to a stereotaxic frame equipped with a heating pad. Anaesthesia was maintained throughout the procedure using continuous delivery of 2% isoflurane through a nose cone at a rate of 1.5 l/min. The scalp was shaved, and a small incision was made along the midline to expose the skull. After levelling the head relative to the stereotaxic frame, the specified injection coordinates were used to mark the locations on the skull directly above each target area and a small hole (0.5 mm diameter) was drilled for each. CTb was delivered through pulled glass micropipettes (inner diameter of tip of approximately 20 μm) using a pressure injection via a micropump (World Precision Instruments). After completing the last injection, the scalp was sutured and mice were administered ketofen (5 mg/kg) to minimize inflammation and discomfort. Mice were recovered from anaesthesia on a heating pad and then returned to their home cage. Mice were euthanized 7 days following injection to allow time for tracer transport, and fresh brain tissue was immediately extracted, embedded in Tissue-Tek O.C.T. Compound (4583, Sakura) and frozen at −80 °C for later cryostat sectioning.

Images of the CTb signal in the injected regions showed that, in the TEa–ECT–PERI injections, the CTb signal covered all cortical layers, whereas in the MOs and SSp injections, the CTb signal appeared relatively weak in L1 and part of L6. Hence, neurons projecting to L1 and L6 of MOs and SSp could be under-represented. In addition, it is known that retrograde tracers such as dye-labelled CTb may not label all neurons projecting to the injected region, and this under-labelling effect could lead to an under-representation of projecting neurons, in particular the double-projecting neurons.

Depending on the location of the injection site, retrograde labelling of TEa–ECT–PERI-projecting neurons in the MOp may display variable patterns[40]. When injection sites are placed in the middle range of the TEa, retrograde labelling in the MOp exhibits a three-layer pattern[57]

staining upper L2/3, upper L5 and L6, whereas injections in the more rostral TEa area leads to less or no L6 labelling. In this work, injection sites were placed in the middle range of TEa that gave the three-layer labelling pattern in the MOp.

## Imaging for CTb-injected tissue

The frozen CTb-injected mouse brain was sectioned the same as described in the 'Tissue preparation for MERFISH' section. A continuous set of 10-μm-thick slices in the region between Bregma approximately 0 and approximately +1.0) was sectioned with approximately every other slice kept and placed onto coverslips for imaging. We used a much higher sampling frequency for CTb-injected samples due to a higher failure rate of this experiment caused by removing the coverslip from the flow chamber after CTb imaging. Tissue slices were immediately fixed by treating with 4% PFA in 1× PBS for 15 min, washed three times with 1× PBS, stained with DAPI and proceed for imaging. As described in the 'MERFISH imaging' section, we used the same imaging buffer, and the sample was first imaged with a low-magnification objective (CFI Plan Apo Lambda ×10, Nikon) for DAPI in a 405-nm channel to produce a low-resolution mosaic of all slices. Next, to align each cell in the tissue with the same tissue slice that would be imaged with the MERFISH probe set later, we picked 10 cells in each coronal slice and recorded the location of the right-side edge for each cell. We then used the mosaic image, created as described above, to locate the MOp region in each slice and generated a grid of FOV positions to cover the MOp region to be imaged. We then switched to the high-magnification objective (CFI Plan Apo Lambda ×60, Nikon) and collected images in the 650-nm channel for CTb-AlexaFluor647, the 560-nm channel for CTb-AlexaFluor555, the 488-nm channel for CTb-AlexaFluor488 and the 405-nm channel for DAPI. We took a single image for each of these channels at the central z-plane.

After the CTb signals were imaged, the sample was removed from the imaging chamber and washed three times by 2× SSC and then permeabilized by 70% ethanol at 4 °C for at least 18 h. The tissue slices were then stained with the same MERFISH probe set, followed by the same MERFISH sample preparation and imaging procedures as described in the 'Tissue preparation for MERFISH' and 'MERFISH imaging' sections. During MERFISH imaging, we first imaged DAPI again with a low-magnification objective, and then located the same 10 cells in each coronal slice that we selected earlier during CTb imaging, and recorded the new location of the right-side edge for each cell. Using the old and new locations of the 10 cells for each slice, we determined the rotation and translation to align the CTb and MERFISH images. Then, MERFISH imaging was performed and the MERFISH images were decoded and segmented as described in the 'MERFISH image analysis and cell segmentation' section. We assigned each cell a projection identity by thresholding the normalized CTb dye intensity for each CTb channel and labelled each cell 'on' or 'off' for each channel. The CTb labelling of the cells were mostly binary (on or off) but still the labelling level varied between cells, therefore the threshold was tuned by manually examining a random subset of the images and was set to a fairly stringent level such that weakly labelled cells were labelled 'off'. The cell-type identities of the CTb-injected samples were determined by training the MERFISH dataset with the MERFISH cell cluster identities without CTb injections using the classifier as described in the 'Correspondence between clusters identified by MERFISH and single-cell sequencing-based measurements' section and predicting on the CTb-injected samples. Each cell in the CTb-injected samples was hence assigned with both a cell-type identity and a projecting-target identity.

## Statistics and reproducibility

Two replicate mice were imaged under each condition. From the two replicate mice imaged for the identification and spatial mapping of cell types, a total of approximately 300,000 cells were imaged, which

generated a sufficient number of single-cell profiles and gave sufficient statistics for the effect sizes of interest. From the two replicate mice imaged for projection target mapping, a total of approximately 190,000 cells were imaged, which gave sufficient statistics for the effect sizes of interest. No statistical methods were used to predetermine sample size. The mice were randomly chosen. For each mouse, the imaging experiments were definitive and no randomization was necessary for this study, hence the experiments were not randomized. The investigators were not blinded to allocation during experiments and outcome assessment because all images were taken under the same condition, and the results were quantitative, which did not require subjective judgement.

The sample sizes for the violin plots in Fig. 1c–e and Extended Data Fig. 3 are listed as follows: Fig. 1c: from left to right, $n$ = 5,585, 6,624, 7,993, 8,373, 5,686, 4,634, 5,431, 2,590, 8,083, 1,830, 2,303, 4,841, 6,570, 1,618, 4,265, 4,267, 5,183, 2,180, 6,699, 1,510, 1,590, 852, 1,417, 538, 2,624, 1,489, 1,810, 4,544, 4,350, 4,189, 3,654, 3,534, 2,009, 1,052, 690, 260, 2,105, 1,244 and 87 cells. Figure 1d: from left to right, $n$ = 504, 161, 475, 480, 403, 259, 146, 154, 150, 124, 137, 391, 343, 241, 257, 123, 96, 222, 299, 154, 125, 200, 379, 648, 555, 462, 338, 414, 137, 152, 1,297, 868, 967, 1,019, 654, 346, 656, 237, 271, 158, 95 and 48 cells. Figure 1e: from left to right, $n$ = 16,013, 2,993, 547, 5,160, 13,223, 5,948, 946, 17,117, 5,435, 3,524, 6,145, 6,888, 295 and 4,339 cells. Extended Data Fig. 3a: upper panel from left to right, $n$ = 2,903, 2,702, 2,873, 4,505, 1,727, 708, 193, 1,258, 3,723, 559, 1,367, 1,648, 3,381, 716, 2,531, 1,134, 3,102, 737, 2,615, 399, 655, 450, 674, 88, 1,233, 606, 808, 1,709, 2,523, 1,715, 1,648, 403, 1,089, 341, 16, 115, 884, 424 and 27 cells; bottom panel from left to right, $n$ = 228, 70, 199, 220, 183, 113, 40, 73, 72, 65, 68, 170, 158, 85, 108, 63, 45, 100, 124, 69, 58, 60, 150, 287, 261, 209, 137, 184, 67, 71, 550, 393, 440, 271, 256, 143, 284, 95, 68, 51 and 47 cells. Extended Data Fig. 3b: upper panel from left to right, $n$ = 405, 1,550, 1,373, 1,793, 863, 235, 135, 140, 1,872, 40, 750, 651, 1,537, 37, 1,253, 637, 1,170, 387, 989, 245, 189, 305, 6, 538, 255, 328, 827, 1,113, 827, 658, 260, 487, 104, 7, 55, 440, 290 and 13 cells; bottom panel from left to right, $n$ = 90, 39, 87, 94, 81, 46, 20, 22, 33, 22, 32, 77, 66, 36, 46, 25, 18, 42, 52, 30, 25, 28, 58, 99, 100, 87, 58, 71, 25, 21, 203, 166, 168, 105, 105, 59, 112, 41, 27, 22 and 24 cells. Extended Data Fig. 3c: upper panel from left to right (ML position = 1/ML position = 6), $n$ = 339/662, 542/365, 352/531, 569/745, 244/315, 39/266, NA/112, 42/221, 537/602, 26/112, 240/108, 161/363, 409/424, 19/123, 251/403, 62/319, 415/254, 39/147, 232/345, NA/69, 82/100, 57/68, 54/103, NA/19, 169/133, 86/105, 111/90, 303/128, 309/228, 173/214, 187/197, 11/193, 122/111, 35/24, 6/NA, 16/13, 135/50 and 32/81 cells; bottom panel from left to right (ML position = 1/ML position = 6), $n$ = 41/100, 314/205, 147/292, 231/339, 107/165, 7/123, NA/88, 15/38, 243/323, 11/NA, 143/54, 43/185, 166/232, 115/199, 37/207, 173/111, 18/90, 72/193, 39/36, 33/35, 29/62, 80/66, 41/47, 48/40, 142/77, 141/132, 73/126, 78/100, 7/135, 47/70, 16/NA, 8/8, 67/38 and 20/63 cells. Violin plots with cell numbers of five or fewer are not shown and the sample size numbers are listed as 'NA' in these cases.

## Reporting summary

Further information on research design is available in the Nature Research Reporting Summary linked to this paper.

## Data availability

The data that support the findings of this study are available from the corresponding author on reasonable request. Raw and processed MERFISH data can be accessed at the Brain Image Library: https://doi.brainimagelibrary.org/ https://doi.org/10.35077/g.21.

## Code availability

The code for the MERFISH image acquisition is available at https://github.com/ZhuangLab. The code for the MERFISH image analysis is available at https://github.com/ZhuangLab/MERlin.

**Acknowledgements** This work was supported in part by the US National Institutes of Health (U19MH114830 to H.Z. and X.Z., and U19MH114821 to H.D. and X.Z.). X.Z. is a Howard Hughes Medical Institute Investigator.

**Author contributions** M.Z., S.W.E., B.Z., Z.Y., H.Z., H.D. and X.Z. designed the experiments. M.Z. and Z.Y. designed the MERFISH gene panel. M.Z. performed the MERFISH experiments. B.Z. performed the CTb tracer injection. M.Z., S.W.E. and K.C. performed the data analysis. M.Z., S.W.E. and X.Z. wrote the paper with input from B.Z., Z.Y., K.C., H.Z. and H.D. X.Z. oversaw the project.

**Competing interests** X.Z. is a co-founder and consultant of Vizgen.

**Additional information**
**Correspondence and requests for materials** should be addressed to X.Z.

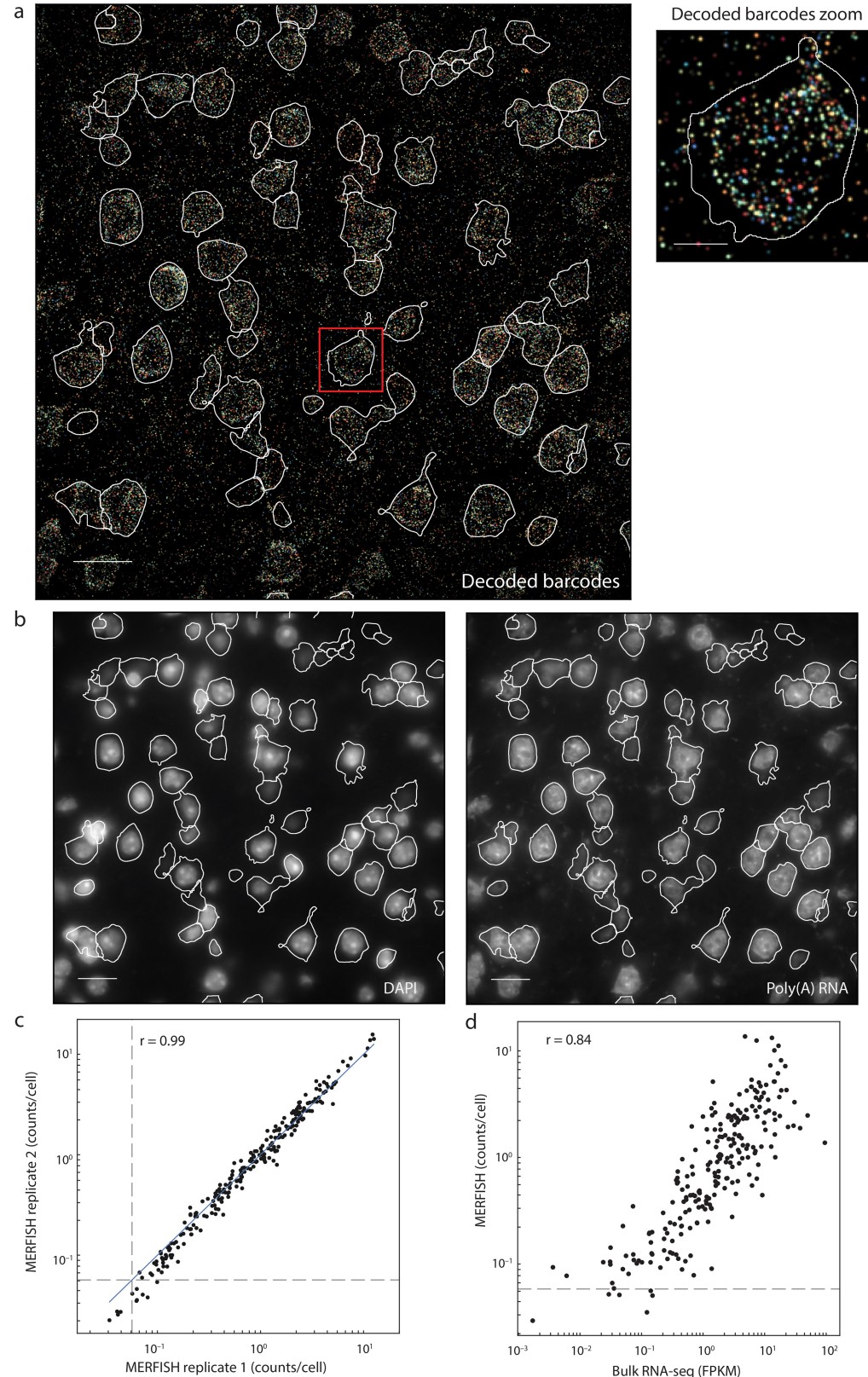

**Extended Data Fig. 1** | See next page for caption.

**Extended Data Fig. 1 | RNA identification and cell segmentation of MERFISH images, replicate reproducibility of MERFISH data, and correlation between MERFISH and bulk RNA-seq results. a**, Decoded MERFISH image of a single field-of-view, shown as a maximum intensity projection across all seven z-planes. In these experiments, we assigned 22-bit Hamming Distance 4, Hamming Weigh 4 barcodes capable of error detection and correction to individual RNA species, and the 22 bits were imaged in 11 rounds of hybridization with two-colour imaging per round. The decoded image shows all pixels that belonged to detected correct barcodes. The pixels were coloured based on their assigned barcodes and the intensity of each pixel was scaled based on the L2-norm of its signal intensity across all bits. Segmented cell boundaries are shown in white. The boxed region of the image is shown at a greater magnification (right). Scale bars, 20 μm (left) and 5 μm (right). **b**, DAPI (left) and poly(A) RNA (right) images for the same field of view as in **a**, with the central z-plane (z = 4.5 μm) shown. These images are used to define the boundaries of each cell, shown in white. Scale bars, 20 μm. **a** and **b** are representative images of more than 5,000 fields of view from two replicate animals. **c**, Scatterplot of the average copy number per cell of individual genes measured by MERFISH for the two replicate animals. The blue solid line indicates equality. The grey dashed lines indicate the average counts per cell of the blank barcodes (that is, valid barcodes that were not assigned to any RNA), which provides an estimate of the false-positive rate. The Pearson correlation coefficient is r = 0.99. **d**, Scatterplot of the average copy number per cell of individual genes determined by MERFISH versus expression level determined by bulk RNA-seq. The dashed line is as defined in **c**. The Pearson correlation coefficient is r = 0.84.

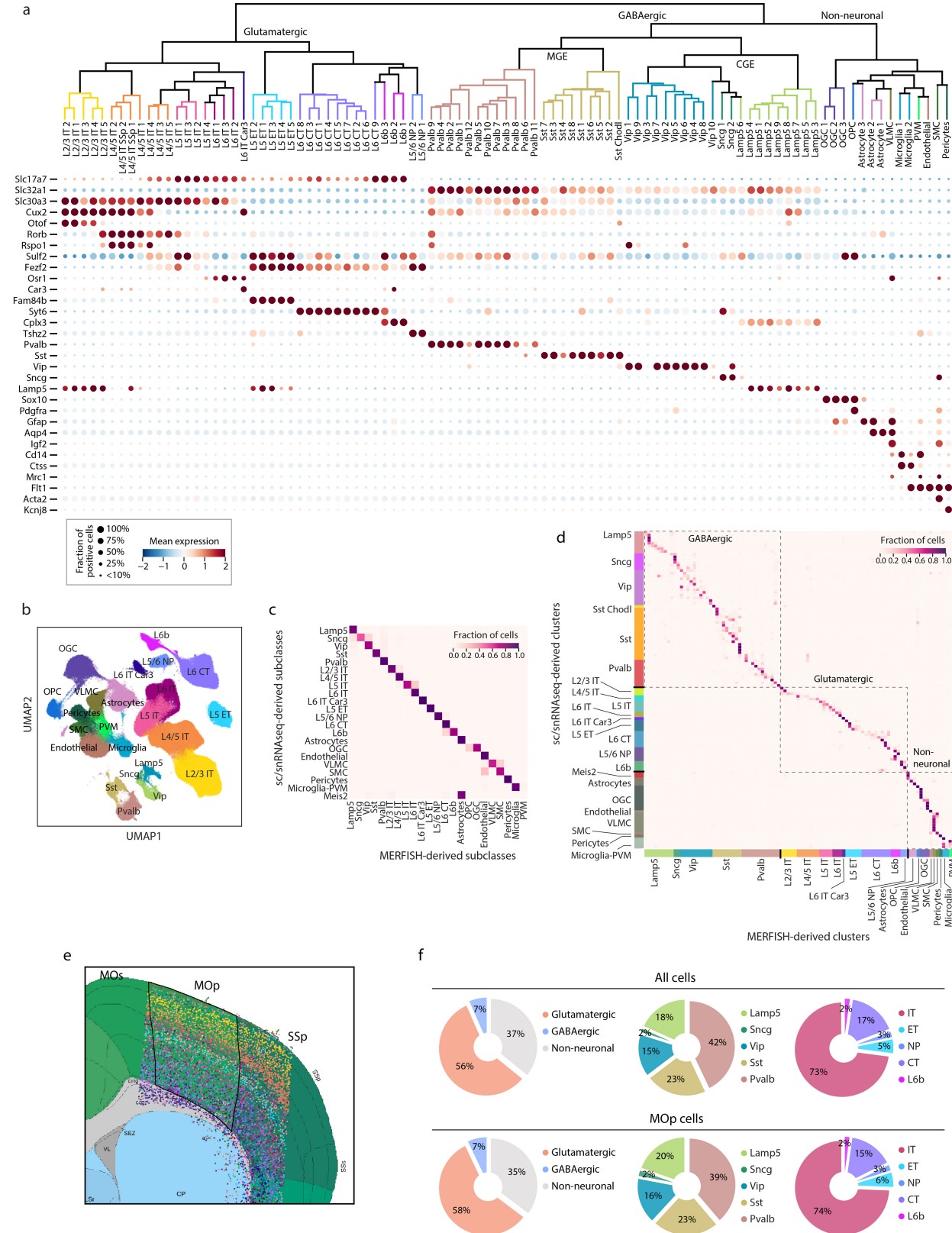

**Extended Data Fig. 2** | See next page for caption.

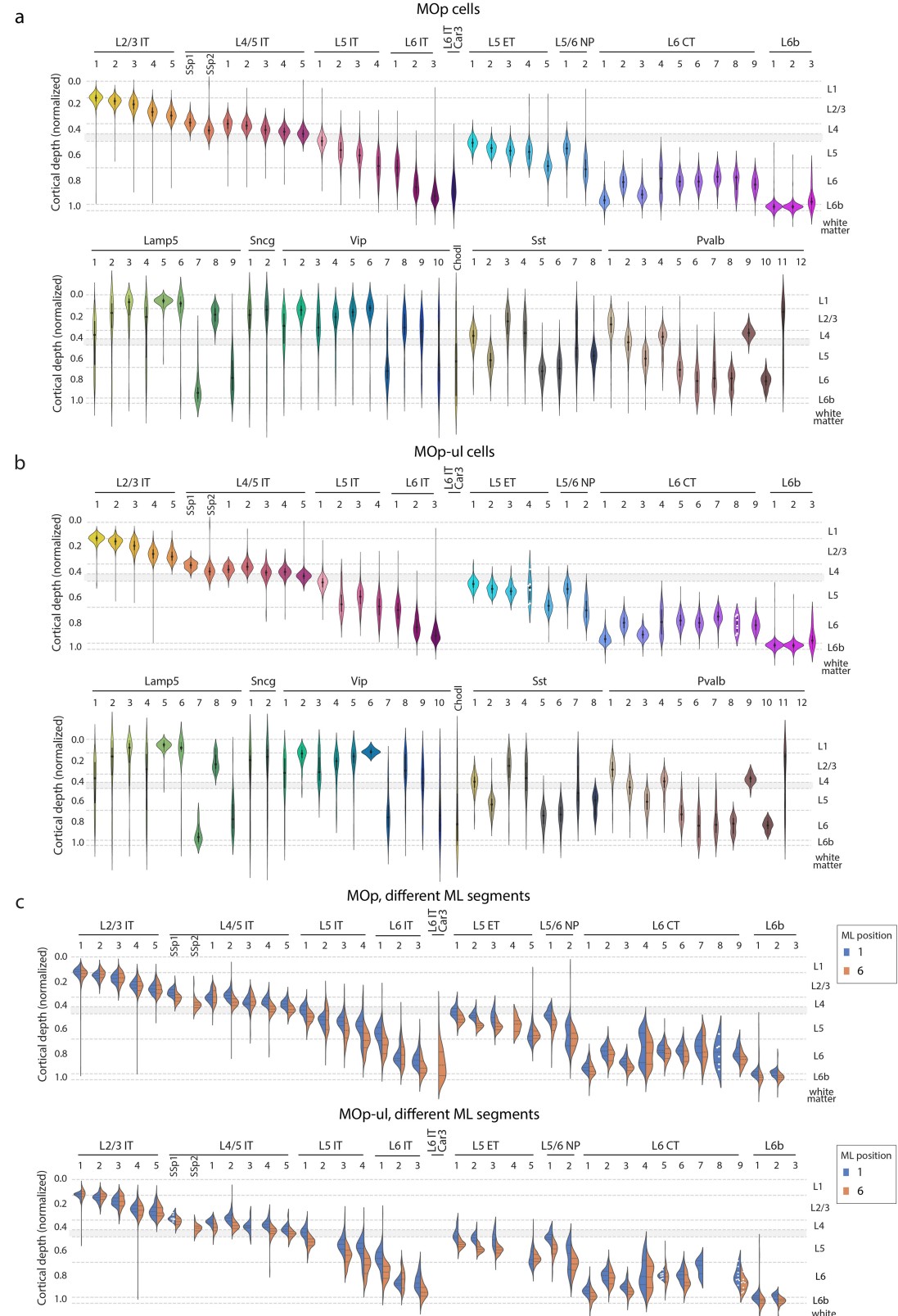

**Extended Data Fig. 2 | Cell-type profiling by MERFISH, registration of MERFISH images to Common Coordinate Framework and the composition of cells. a**, Dendrogram showing the hierarchical relationship among the 39 glutamatergic, 42 GABAergic, and 14 non-neuronal clusters identified by MERFISH, constructed based on the z-scored cluster expression profiles and coloured by the subclass that each cluster belongs to (top). Expression of markers genes for the subclasses are also displayed (bottom). **b**, UMAP of cells measured by MERFISH coloured based on the subclasses of cells. **c**, Correspondence between the subclasses of cells determined by MERFISH and the subclasses determined by scRNA-seq and snRNA-seq datasets generated in a companion study[23]. A neural-net classifier was used to predict a MERFISH cluster label for each cell in the snRNA-seq 10x v3 B dataset[23]. The fraction of cells from any given subclass identified by scRNA-seq and snRNA-seq that was predicted to have each MERFISH subclass label was plotted. **d**, Correspondence between clusters identified by MERFISH and by scRNA-seq and snRNA-seq. As in **c**, but with the fraction of cells from any given cluster identified by scRNA-seq and snRNA-seq that was predicted to have each MERFISH cluster label plotted. Each row or column corresponds to a cluster and the coloured bars on the left and bottom mark the subclasses. **e**, An example MERFISH-derived cell-type spatial map overlaid on the Allen CCF v3 with cells coloured by their cluster identities as in Fig. 1a. The CCF v3 image is from the Allen Brain Atlas (http://atlas.brain-map.org/; credit: Allen Institute). The boundaries of the MOp are shown in black. We note that the MOp–MOs and MOp–SSp borders, as well as all other cortical area borders in the Allen CCF v3, are largely perpendicular to the cortical surface in the 3D space. The reason that they do not always appear perpendicular to the cortical surface line in the coronal sections, as shown here, is because the coronal sections themselves are not always perpendicular to the cortical surface (see more details about this effect at: https://community.brain-map.org/t/ccfv3-highlights-tilting-at-the-cortex/1000; credit: Allen Institute). **f**, Top: fractions of cells in the entire imaged region that belong to each of the three major cell classes (glutamatergic, GABAergic and non-neuronal) (left). Fractions of GABAergic cells in the entire imaged region that belong to each of the GABAergic subclasses (middle). Fractions of glutamatergic cells in the entire imaged region that belong to each of the glutamatergic subclasses (right). The mean fractions are shown and the 95% confidence intervals are less than 0.3%. Bottom: same as the top panel but for cells in the MOp. The mean fractions are shown and the 95% confidence intervals are less than 0.4%.

**Extended Data Fig. 3 |** See next page for caption.

**Extended Data Fig. 3 | Cortical depth distributions of neuronal cell clusters in the MOp and an approximate MOp upper limb region. a**, **b**, As in Fig. 1c, d but for glutamatergic and GABAergic clusters in the MOp (**a**) and an approximate MOp upper limb (MOp-ul) region (**b**). We selected the region between Bregma 0 and +1.0 within the MOp as an approximation for the MOp-ul region based on previous literature[30] and a companion paper[31]. This region is considered as the primary part of the MOp-ul because it contains the densest pyramidal neurons that directly project to the intermediate horn and ventral horn of the cervical spinal cord and, in the meantime, shows minimal projection to the lower limb[31]. In the violin plots, the black dot represents the median, the thick black bar represents the interquartile range, and the edges define minima and maxima. **c**, Effect of the layer-thickness variations along the medial–lateral (ML) direction on the cortical depth distributions of the neuronal clusters. Top: we divided the MOp into six segments along the ML direction, each covering a narrow ML range such that the layer-thickness variations within each segment are negligible. We then determined the cortical depth distributions of the neuronal clusters in each of the six ML segments and display those for the most medial (blue; ML position 1) and most lateral (orange; ML position 6) segments. Most of the clusters showed only a modest difference in their cortical depth distributions between different ML segments, with a small number of exceptions (such as L4/5 IT SSp 1 and 2, L5 ET 4, L6 CT 8 and L6 IT Car 3). These exceptions showed large differences due to the region-dependent presence of these clusters. Only the distributions of the glutamatergic clusters are shown here because the relatively low abundance of the GABAergic neurons makes the comparison of their distributions in different ML segments statistically less sound. Bottom: as in the top panel but for glutamatergic clusters in the approximate MOp-ul region as defined in **b**. In the violin plots in **c**, the centre dashed line represents the median, the other two dashed lines represent the interquartile range, and the edges define minima and maxima. For all violin plots in **a**–**c**, the clusters with cell numbers of five or fewer are not shown, and the clusters with cell numbers of ten or fewer (but more than five) are shown with individual data points as white dots.

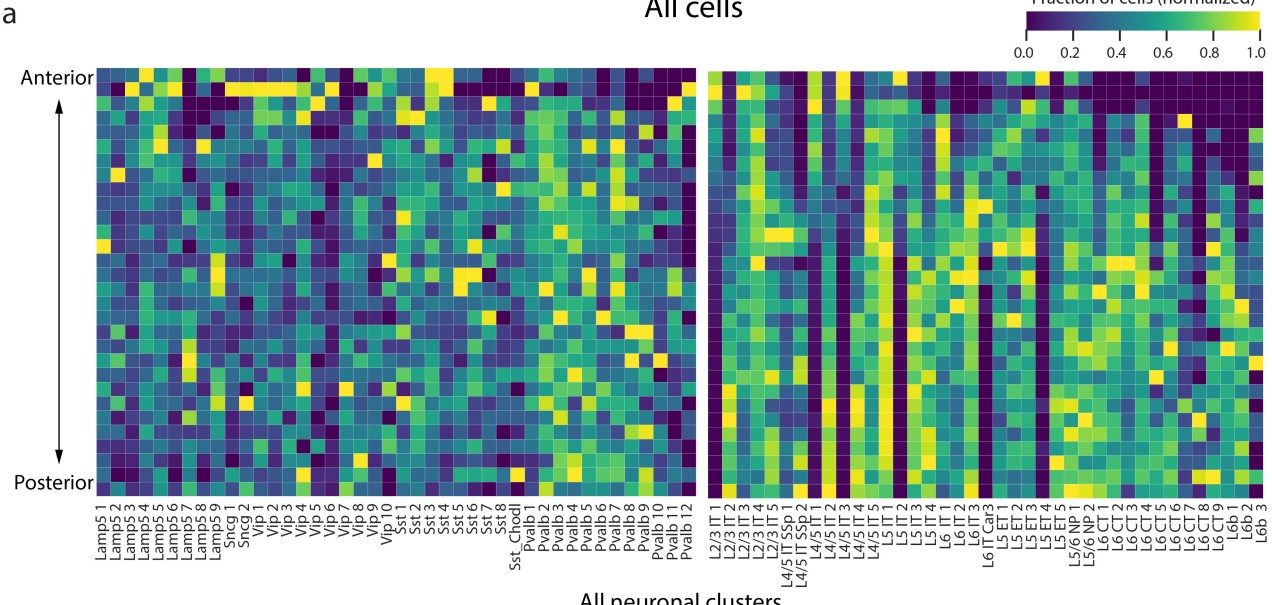

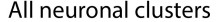

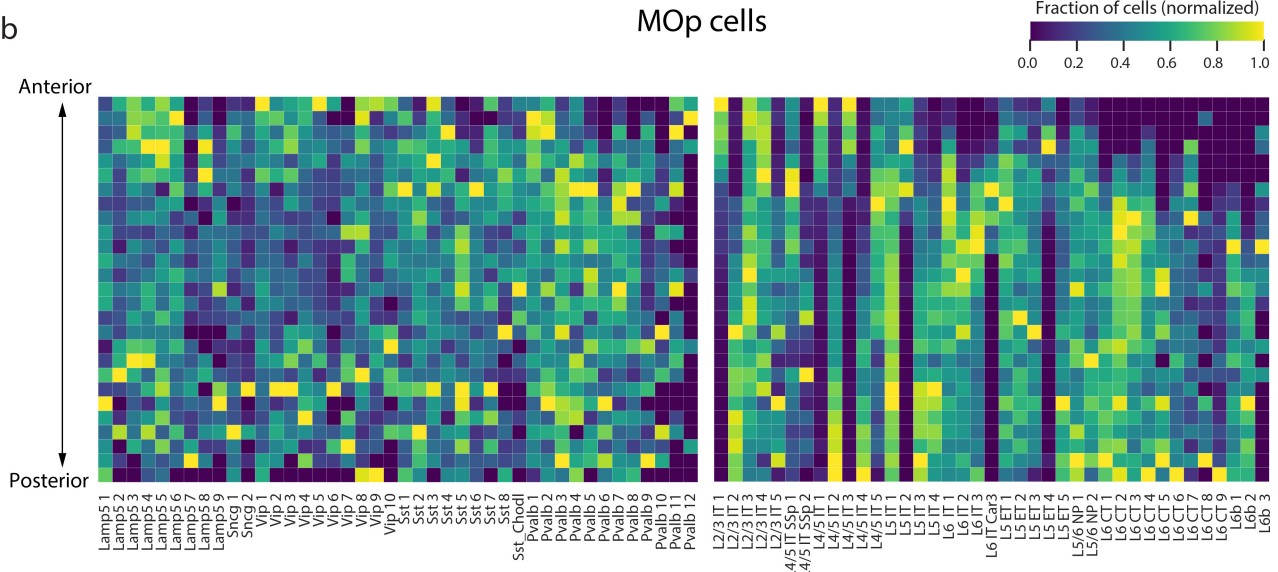

**Extended Data Fig. 4 | Anterior–posterior distribution of neuronal clusters. a**, Heatmap quantifying the anterior–posterior distribution of the neuronal clusters in the entire imaged region including the MOp and its adjacent areas. Slices were arranged from anterior-most to posterior-most based on their Bregma coordinates (Bregma +2.5 to −0.8). For each cluster, the fraction of cells found in each slice was determined and normalized to the maximum across all slices. **b**, As in **a** but for the neurons within the MOp.

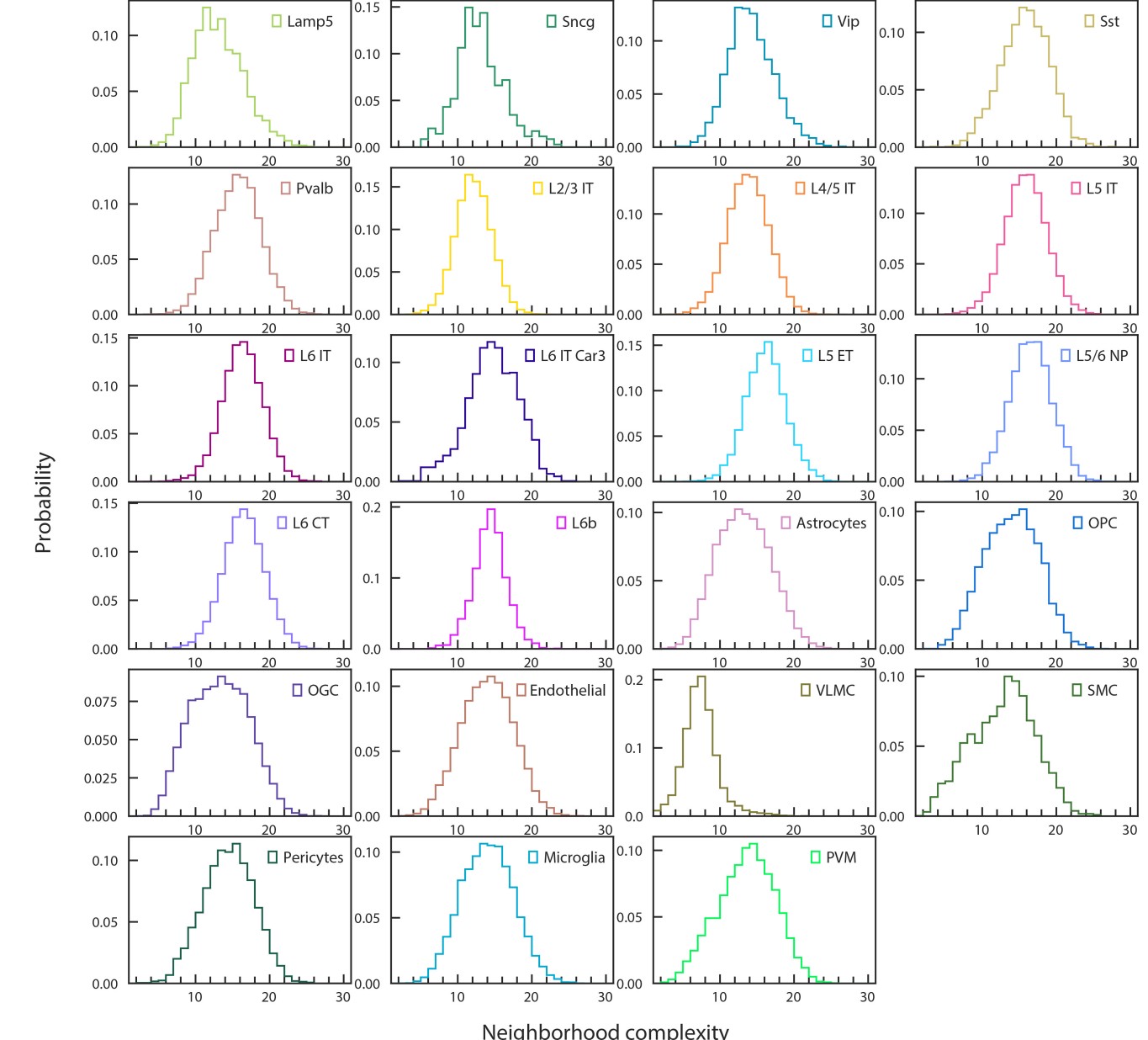

**Extended Data Fig. 5 | Neighbourhood complexity of individual cells belonging to different subclasses.** The neighbourhood complexity of a cell is defined as in Fig. 1f. A normalized histogram of the neighbourhood complexity for all cells from a given subclass is shown for each cell subclass.

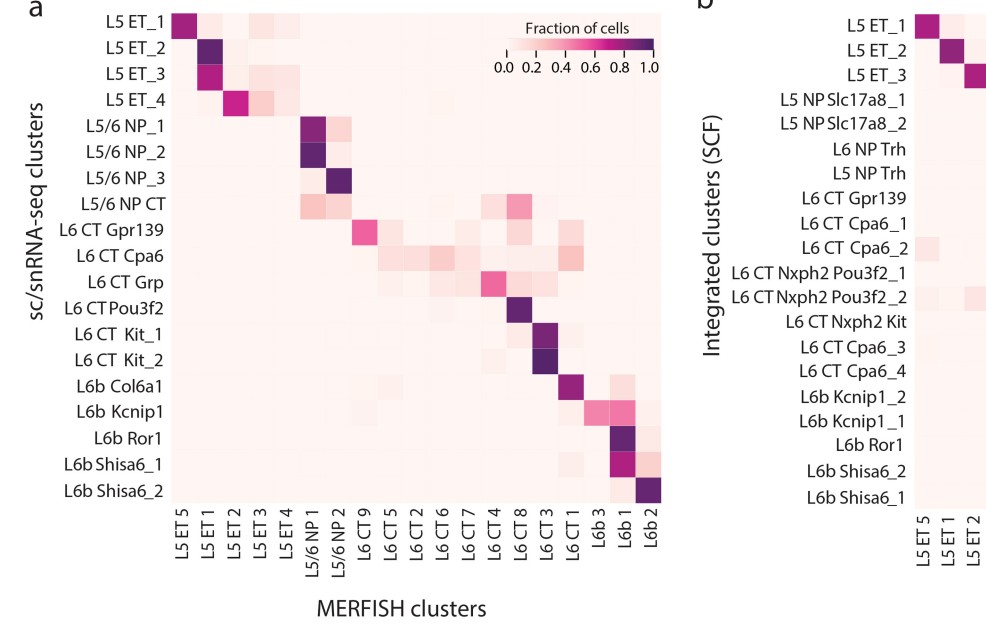

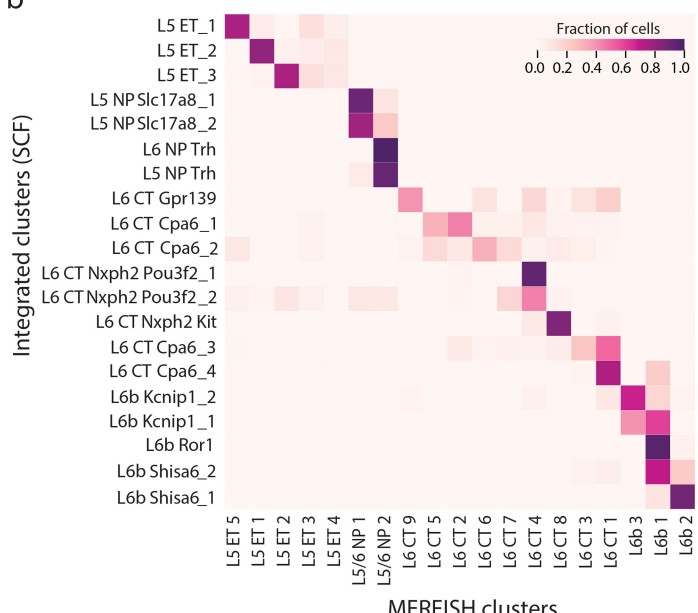

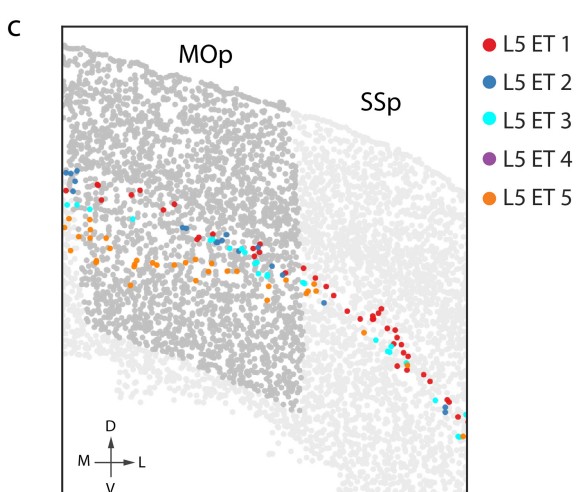

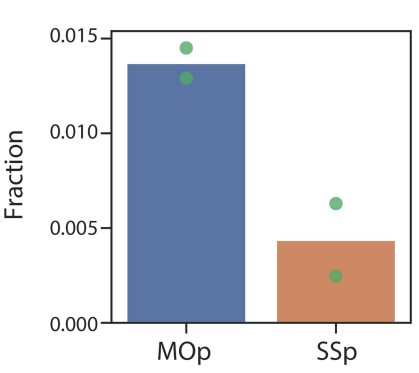

**Extended Data Fig. 6 | Additional analyses of the L5 ET, L5/6 NP, L6 CT and L6b clusters. a**, Correspondence between the L5 ET, L5/6 NP, L6 CT and L6b clusters determined by MERFISH and those identified by scRNA-seq and snRNA-seq. **b**, Correspondence between the L5 ET, L6 CT, L5/6 NP and L6b clusters determined by MERFISH and those identified by integrated analysis of scRNA-seq, snRNA-seq, snATAC-seq and snmC-Seq datasets using SingleCellFusion[23]. The classifier approach used in **a** and **b** to determine the correspondence is as described in Extended Data Fig. 2c. **c**, Left: a coronal slice

(Bregma approximately +0.7) highlighting the L5 ET cells coloured by cell clusters in the MOp and the neighbouring SSp region. Cells other than the L5 ET cells are shown in dark grey in the MOp to highlight the MOp region, and cells other than the L5 ET cells outside the MOp are shown in light grey. Scale bars, 200 μm. Right: comparison of the abundance of L5 ET 5 neurons in the MOp and SSp. The fraction of L5 ET 5 cells with respect to the total cell number detected in the MOp or SSp are shown. $n = 2$ replicate animals, with individual data points from each animal shown.

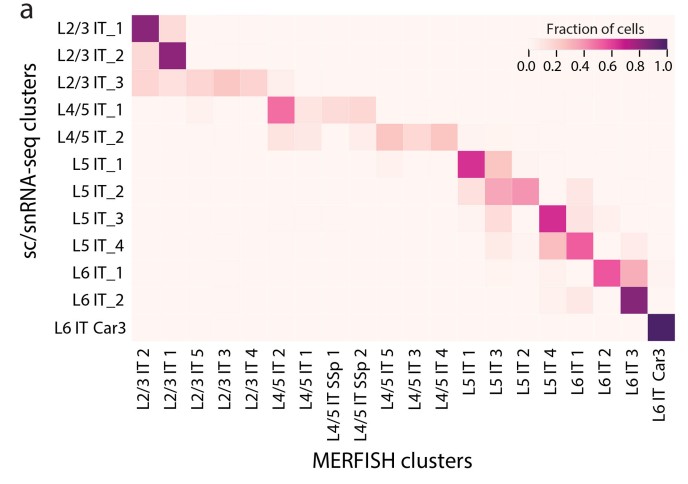

**a**

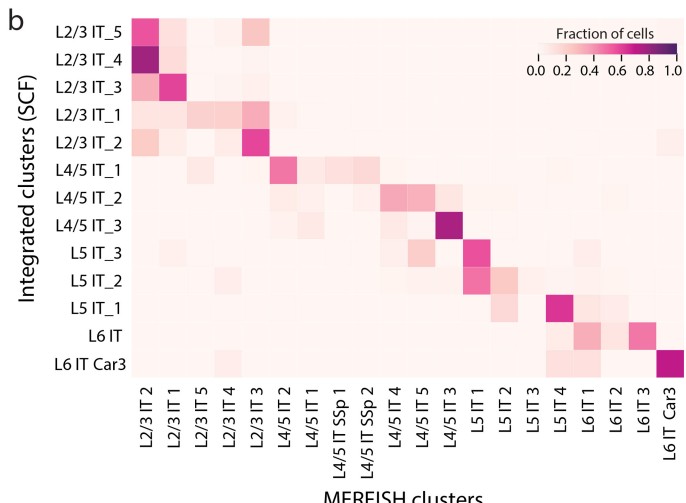

**b**

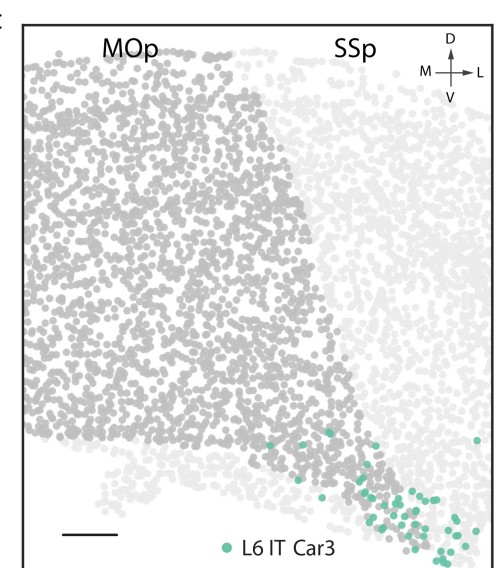

**c**

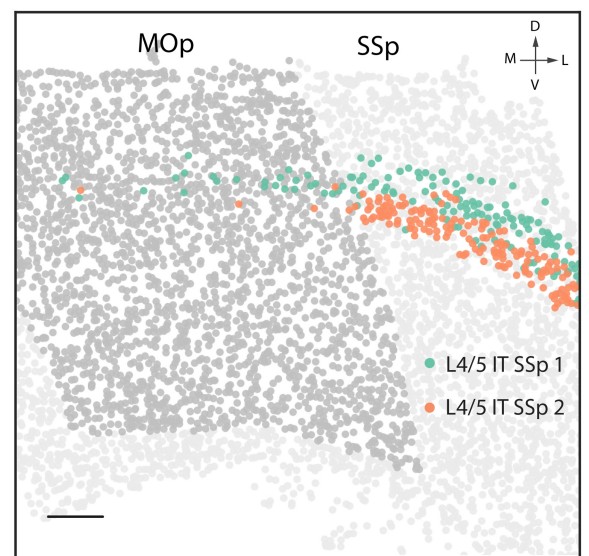

**d**

**Extended Data Fig. 7 | Correspondence between the MERFISH IT clusters and the IT clusters determined by scRNA-seq and snRNA-seq analysis and by integrated analysis, and spatial distributions of L6 IT Car3 and L4/5 IT SSp 1 and 2 clusters. a**, Correspondence between the IT clusters identified by MERFISH and those identified by scRNA-seq and snRNA-seq. **b**, Correspondence between the IT clusters identified by MERFISH and those identified by the integrated clustering analysis of scRNA-seq, snRNA-seq, snATAC-seq and snmC-seq using SingleCellFusion[23]. The correspondence in

**a** and **b** is determined using a classifier approach as described in Extended Data Fig. 2c. **c**, A coronal slice (Bregma approximately +1.1) highlighting the L6 IT *Car3* cluster (green). **d**, A coronal slice (Bregma approximately +0.9) highlighting the L4/5 IT SSp 1 (green) and L4/5 IT SSp 2 (orange) clusters. In both **c** and **d**, all other cells within the MOp are coloured in dark grey to highlight the MOp region, and all other cells outside the MOp are shown in light grey. Scale bars, 200 μm (**c**, **d**).

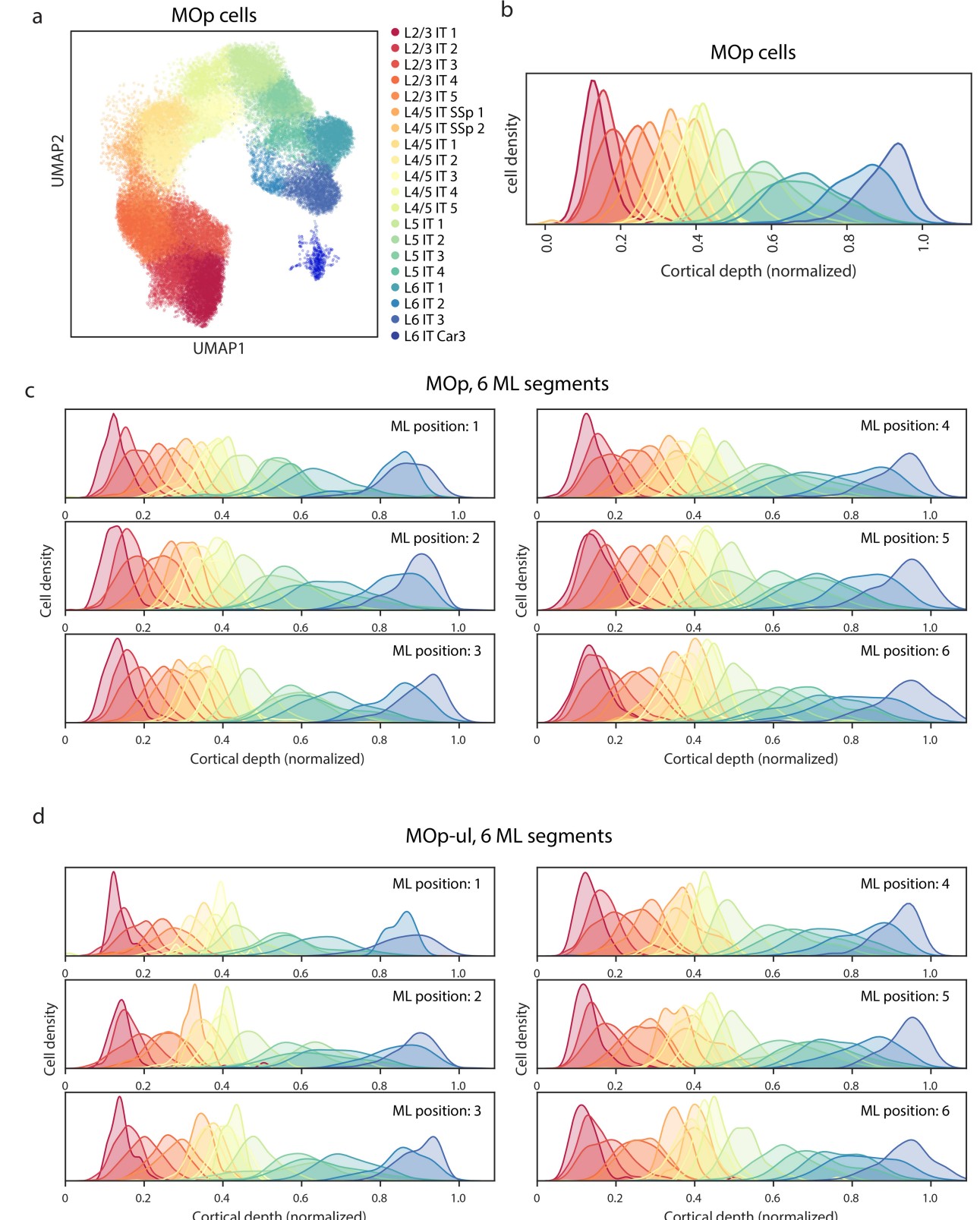

**Extended Data Fig. 8** | See next page for caption.

**Extended Data Fig. 8 | Gene expression profiles and cortical depth distributions of IT cell clusters in the MOp. a**, UMAP of the IT clusters for cells in the MOp. **b**, Cortical depth distributions of IT clusters in the MOp region, with individual IT clusters coloured as in **a. c**, Cortical depth distributions of IT clusters in different medial–lateral (ML) segments of the MOp region. The IT clusters are coloured as in **a**. As the variation in layer thicknesses along the ML direction could broaden the cortical depth distributions of the clusters, to assess whether the spatial overlap between different IT clusters could be caused by this effect, we divided the MOp into six segments along the ML direction, each covering a narrow ML range such that the layer-thickness variations within each ML segment are negligible. We then determined the cortical depth distributions of the IT clusters in each of the six ML segments. The spatial overlap between the IT clusters was still observed in each of the six ML segments. **d**, Cortical depth distributions of IT clusters in different ML segments in the approximate MOp upper limb region (between Bregma 0 and +1.0, as defined in Extended Data Fig. 3b). The spatial overlap between the IT clusters was still observed in each of the six ML segments in this region. Clusters with a low cell number (five or fewer) found in any individual ML segments are not shown.

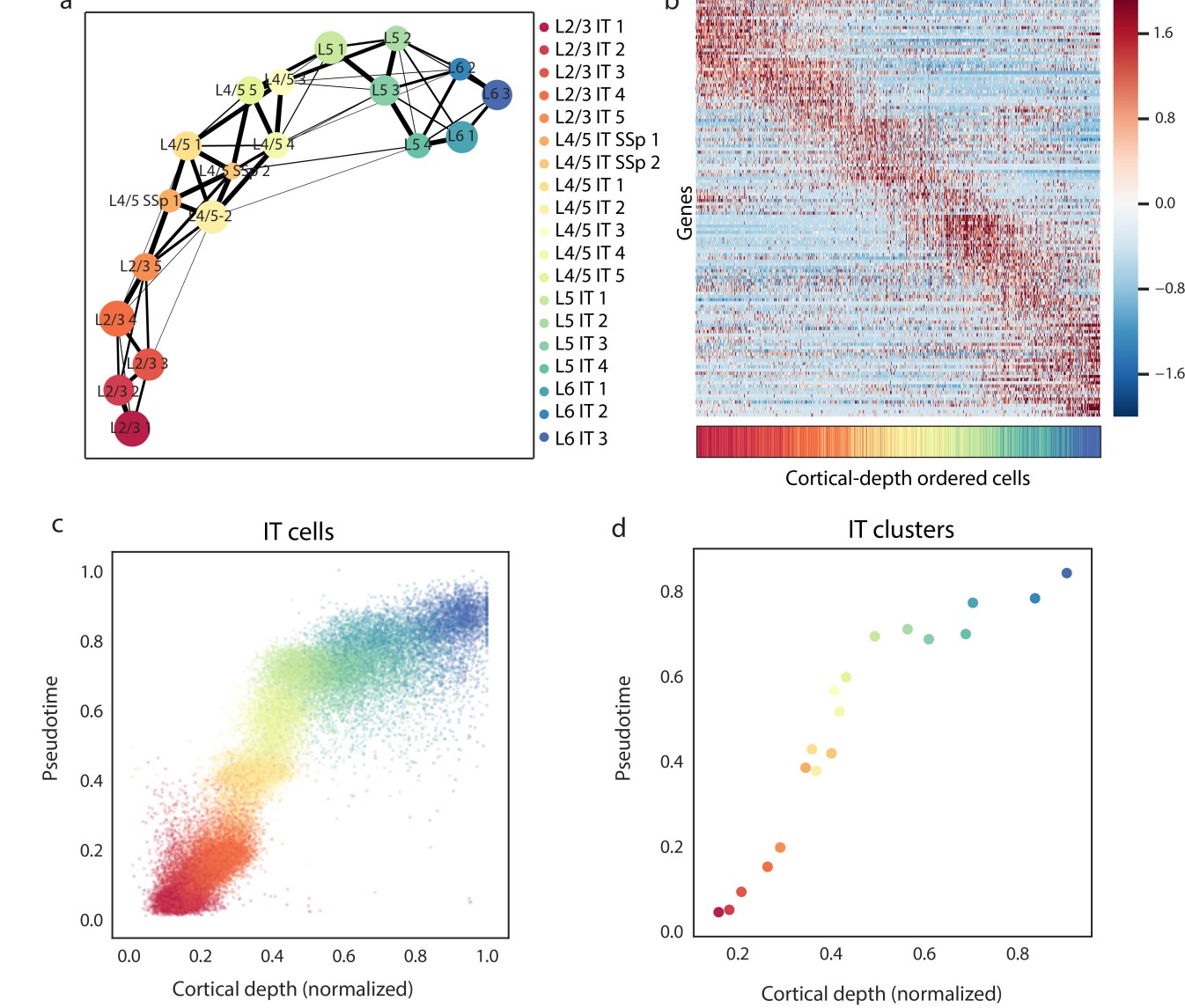

**Extended Data Fig. 9 | Correlated gradients in gene expression and cortical depth across IT neurons in the MOp. a**, Same as Fig. 3d, but for IT neurons in the MOp. **b**, Same as Fig. 3e, but for differentially expressed genes of all IT neurons within the MOp across cortical depth. **c**, **d**, Same as Fig. 3f, but for individual IT cells (**c**) and individual IT clusters (**d**) in the MOp.

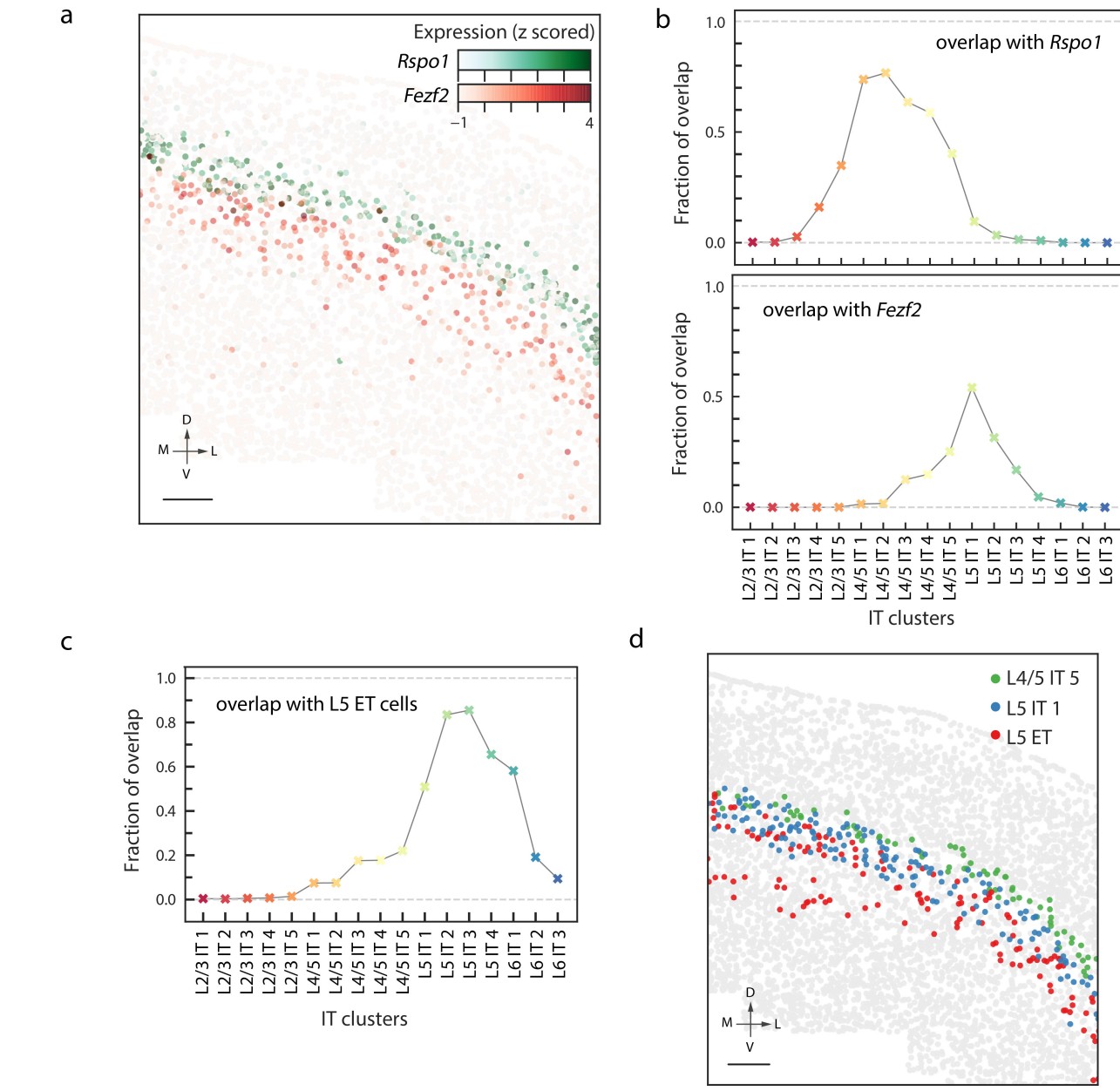

**Extended Data Fig. 10 | Spatial overlap between individual IT clusters and the L4 marker gene *Rspo1*, the L5 marker gene *Fezf2* and L5 ET cells.** **a**, A coronal slice (Bregma approximately +1.0) highlighting the IT cells that express *Rspo1* (green) and *Fezf2* (red). Scale bar, 200 μm. **b**, Spatial overlap between individual IT clusters and *Rspo1* (top) and *Fezf2* (bottom). **c**, Spatial overlap between individual IT clusters and L5 ET cells. **d**, The same coronal slice as in **a**, but highlighting the L4/5 IT 5 (green), L5 IT 1 (blue) and L5 ET (red) cells. Scale bar, 200 μm. See Methods ('Layer boundary assessment' section) for how the spatial overlap is determined.

# Reporting Summary

Nature Research wishes to improve the reproducibility of the work that we publish. This form provides structure for consistency and transparency in reporting. For further information on Nature Research policies, see Authors & Referees and the Editorial Policy Checklist.

## Statistics

For all statistical analyses, confirm that the following items are present in the figure legend, table legend, main text, or Methods section.

| n/a | Confirmed | |
|---|---|---|
| ☐ | ☒ | The exact sample size (*n*) for each experimental group/condition, given as a discrete number and unit of measurement |
| ☐ | ☒ | A statement on whether measurements were taken from distinct samples or whether the same sample was measured repeatedly |
| ☒ | ☐ | The statistical test(s) used AND whether they are one- or two-sided *Only common tests should be described solely by name; describe more complex techniques in the Methods section.* |
| ☒ | ☐ | A description of all covariates tested |
| ☒ | ☐ | A description of any assumptions or corrections, such as tests of normality and adjustment for multiple comparisons |
| ☐ | ☒ | A full description of the statistical parameters including central tendency (e.g. means) or other basic estimates (e.g. regression coefficient) AND variation (e.g. standard deviation) or associated estimates of uncertainty (e.g. confidence intervals) |
| ☒ | ☐ | For null hypothesis testing, the test statistic (e.g. *F*, *t*, *r*) with confidence intervals, effect sizes, degrees of freedom and *P* value noted *Give P values as exact values whenever suitable.* |
| ☒ | ☐ | For Bayesian analysis, information on the choice of priors and Markov chain Monte Carlo settings |
| ☒ | ☐ | For hierarchical and complex designs, identification of the appropriate level for tests and full reporting of outcomes |
| ☐ | ☒ | Estimates of effect sizes (e.g. Cohen's *d*, Pearson's *r*), indicating how they were calculated |

*Our web collection on statistics for biologists contains articles on many of the points above.*

## Software and code

Policy information about availability of computer code

| Data collection | MERFISH imaging data was collected using custom Python code to control the microscope. This code is available at https://github.com/ZhuangLab. |
|---|---|
| Data analysis | The MERFISH data was analyzed using custom Python code. This code is available at https://github.com/ZhuangLab/MERlin. Other packages used in data analyses include: Scanpy (version 1.4); Bioconductor limma (version 3.38); Scrublet (version 0.2). |

For manuscripts utilizing custom algorithms or software that are central to the research but not yet described in published literature, software must be made available to editors/reviewers. We strongly encourage code deposition in a community repository (e.g. GitHub). See the Nature Research guidelines for submitting code & software for further information.

## Data

Policy information about availability of data

All manuscripts must include a data availability statement. This statement should provide the following information, where applicable:

- Accession codes, unique identifiers, or web links for publicly available datasets
- A list of figures that have associated raw data
- A description of any restrictions on data availability

Data availability statement is included in the manuscript, which states:
The data that support the findings of this study are available from the corresponding author upon reasonable request. Raw and processed MERFISH data can be accessed at the Brain Image Library : https://doi.brainimagelibrary.org/doi/10.35077/g.21.

# Field-specific reporting

Please select the one below that is the best fit for your research. If you are not sure, read the appropriate sections before making your selection.

☒ Life sciences ☐ Behavioural & social sciences ☐ Ecological, evolutionary & environmental sciences

For a reference copy of the document with all sections, see nature.com/documents/nr-reporting-summary-flat.pdf

# Life sciences study design

All studies must disclose on these points even when the disclosure is negative.

| | |
|---|---|
| Sample size | Two replicate animals were imaged under each condition. From the two replicate animals imaged for the identification and spatial mapping of cell types, a total of ~300,000 cells were imaged, which generated a sufficient number of single-cell profiles and gave sufficient statistics for the effect sizes of interest. From the two replicate animals imaged for projection target mapping, a total of ~190,000 cells were imaged, which gave sufficient statistics for the effect sizes of interest. |
| Data exclusions | We did not exclude any data from consideration. All images were included in the primary analysis. |
| Replication | Reported results were replicated from two animals under each condition. |
| Randomization | Two male animals were randomly chosen for the identification and spatial mapping of cell types, and two male animals were randomly chosen for the projection pattern study. For each animal, the imaging experiments were definitive and no randomization was necessary for this study. |
| Blinding | Blinding during collection was not needed because all images were taken under same condition. Blinding during analysis was not necessary because the results were quantitative and did not require subjective judgment. Blinding is not typically used in the field. |

# Reporting for specific materials, systems and methods

We require information from authors about some types of materials, experimental systems and methods used in many studies. Here, indicate whether each material, system or method listed is relevant to your study. If you are not sure if a list item applies to your research, read the appropriate section before selecting a response.

### Materials & experimental systems

| n/a | Involved in the study |
|---|---|
| ☒ ☐ | Antibodies |
| ☒ ☐ | Eukaryotic cell lines |
| ☒ ☐ | Palaeontology |
| ☐ ☒ | Animals and other organisms |
| ☒ ☐ | Human research participants |
| ☒ ☐ | Clinical data |

### Methods

| n/a | Involved in the study |
|---|---|
| ☒ ☐ | ChIP-seq |
| ☒ ☐ | Flow cytometry |
| ☒ ☐ | MRI-based neuroimaging |

# Animals and other organisms

Policy information about studies involving animals; ARRIVE guidelines recommended for reporting animal research

| | |
|---|---|
| Laboratory animals | Adult C57BL/6J male mice aged 57-63 days were used in this study. Mice were maintained on a 12h/12h light/dark cycle, at a temperature of 22 ± 1 °C, a humidity of 30-70%, with ad libitum access to food and water. |
| Wild animals | The study did not involve wild animals. |
| Field-collected samples | The study did not involve samples collected from the field. |
| Ethics oversight | Harvard University Institutional Animal Care and Use Committee; University of South California Institutional Animal Care and Use Committee. |

Note that full information on the approval of the study protocol must also be provided in the manuscript.

