## [Peer Review File · Nature]

Manuscript Title: Spatially resolved cell atlas of mouse primary motor cortex by MERFISH

Reviewer Comments & Author Rebuttals

Reviewer Reports on the Initial Version:

Referees' comments:

Referee #1 (Remarks to the Author):

This study, representing the spatial transcriptomics arm of the BICCN effort to characterize cortical cells in mouse "MOp", uses MERFISH to assess gene expression as a function of soma locations across layers. This work has potentially high significance for both motor cortex research and for understanding cortical organization across areas and species. Findings include: (i) evidence against conventional layers playing the primary role as a determinant of cellular identity; (ii) evidence against molecular specification playing a role in the identity of different projectionally defined subtypes of IT neurons projecting to other cortical areas in various patterns (p.12); (iii) an inventory of gene expression of (nearly) all known cell types (importantly including nonneuronal classes) as a function of soma location, mainly in terms of depth from cortex, but also A-P position within MOp; (iv) quantitative information about relative abundances of cell types; (v) further evidence for and characterization of various subtypes of neurons, particularly interneurons, which are relatively under-studied in motor and frontal areas. Thus there are a number of important results. Comments, concerns, and suggestions are as follows.

Spatial analysis: How exactly were the spatial analyses performed? How was soma depth and all the related morphometric parameters (depth of cortex, etc) actually measured? Was the 'sloping' of layers (well known in MOp, particularly in the medial part, and clearly evident in the figures) and variation in cortical thickness taken into account and compensated for in the analyses? How were layers defined and identified? More on these points below.

Sloping layers: Did the sloping of the layers in MOp contribute to blurring of what are actually sharper laminar distributions? E.g. in Fig 3, the distributions of the labeled cells are very different on the far left vs the middle and right of the section; were the distributions in the plot on the right artifactually broadened by pooling these together? This consideration is a general one, applying to all the depth-of-soma analyses. In the extreme, does the "continuous spectrum" of the IT gene expression to some extent reflect artifactual 'smearing' of laminar distributions?

"MOp": What is this region, exactly? I understand that it's the Allen Atlas's parcellation, but what's it based on? Why is the border between MOp and MOs drawn where it is? It doesn't seem to be based on any clear functional or even anatomical considerations. The article could potentially be strengthened by focusing (mainly, or at least in a sub-analysis) on the forelimb region within MOp. Currently, inclusion of the most anterior, lateral, and medial regions of "MOp", which don't seem to be well-defined as "primary" motor areas, makes it hard to interpret the findings in terms of their significance for "primary" motor cortex. The authors may wish to point out that "MOp", as defined here, is not necessarily equivalent to "primary motor cortex", at least not as it's most commonly defined in the literature (e.g. based on ICMS thresholds, presence of corticospinal neurons, functional imaging/electrophysiology, etc.).

"L5": What exactly is this, and how does it relate to the much more precise designation of a distinct L5A and L5B, which is often used in studies of mouse motor, frontal, and somatosensory cortex? I understand that these layers are indistinct in some regions of the mouse cortex where

layers appear merged (e.g. cingulate cortex, occipital areas), but in most areas and species there is a clear distinction between L5A and L5B, and this is certainly the case for mouse M1. Up to the authors, but I would recommend the following laminar definitions: L1: easily defined (e.g. absence of PyN); L2/3: everything between L1 and L4; L4: RORB labeling; L5A: ETV1 and absence of PT neurons; L5B: presence of PT neurons; L6A: presence of CT neurons; L6B: as you have done. For example, in Extended Data Figure 6 the full laminar distribution of the PT neurons defines L5B (which is much broader in MOp than SSp).

Nomenclature: Why introduce the new and much less precise term "ET" to replace the well-established "PT" nomenclature? To be sure, "PT" is an imperfect term – as are "neocortex", "isocortex", and many other everyday terms in cortical neuroanatomy. Yet "PT" is widely used and hardly problematic especially in motor cortex research, where it's been the preferred term for over half a century. If you really want to replace it, there's the equivalent and more technically precise (but more of a mouthful) term "subcerebral projection neurons". The problems with "ET" are multiple, and hardly worth going into, but in a nutshell: it implies that these neurons don't form IT projections (they do), and that they alone form ET projections (not so, as CT neurons do, too), and it tries (erroneously in this case) to define something not by what it is but by what it's not ("not-IT").

Minor:

Taxonomic naming conventions: order should go from major to minor. E.g. IT-L2/3-subtype, not L2/3-IT-subtype1, etc. For PT and CT, no need to indicate layer, since they are by definition in one particular layer, but can indicate subtype/sublayer as appropriate (e.g. PT-subtype1, PT-subtype2, etc).

Corticospinal neurons: The archetypal cell type of primary motor cortex, yet never mentioned (in what is intended as a "comprehensive" analysis of MOp cell types). Consider at least a brief mention, e.g. with citation to the closely related Bakken et al. paper (ref. 59). Could point out that there's no unique molecular marker for corticospinal neurons.

p.3, "Single-cell gene expression..." section, first paragraph – this is basically a description of the recipe for generating the gene panel. But what's the rationale for the particular blend of the three subsets of genes? Surely much thought went into this, so it would be nice to know the thinking behind it. Also, can citations be provided for the "prior knowledge"?

p.5 – cell class abundances are given as percentages; what are the 95% CI?

Sometimes unclear use of the term "layer" – e.g. p.6, does "layers" refer to one of the supposed "6 layers", or to "strata" or perhaps "sublayers"?

p.7 – the "high level of local cellular heterogeneity" – is this reflective of MOp in particular? Or similar to the "salt-and-pepper" organization of mouse V1? Is it a mouse thing in general?

p.7 – "L5 ET 4 mostly resided outside of MOp" – where, exactly?

p.8 – "unique medulla-projecting L5 ET 5 cell type was mainly present in MOp but rarely in the adjacent SSp region" – this is surprising; suggests that there different types of medulla-projecting PT cells in MOp and S1? Cf. Oberlaender papers.

p.9 – "it is also possible to impute marker genes ... which would ... provide a more complete marker gene sets for the MERFISH clusters" – and would give better nomenclature; i.e., gene names instead of subtype numbering (1, 2, etc.), which is hard to remember, confusing, and will likely need revision anyways. Since this is intended as the definitive study (or set of studies), isn't now the time to identify the marker genes and use them for naming clusters/subtypes?

p.9 – IT section: is some depth used for the clustering analysis? and/or layer? or is the clustering done agnostic to soma depth, and the clusters turn out to have laminar distributions? This should be clarified at the outset. Also, how was classification done? By stipulating the number of layers first? How would it differ if layer 4, 5A, 5B were considered separate layers?

p.11 – choice of “MOs” and “SSp” makes sense but “TEa” seems like an odd choice, compared to for example S2, which has relatively well-characterized interconnections with M1 and S1. Moreover, how cleanly can TEa be injected using a vertical (dorsal) approach as seems to have been used here? And, wouldn’t such an approach result in labeling at only one laminar level, which could affect the results if axons ramifying in other layers went unlabeled?

p.11 – “... enriched in upper layers L2/3 and L4/5; ... enrichment in upper L2/3, deep L4/5 and L6” – this sentence exemplifies the problems with the lack of recognition of L5A – previous studies have described labeling of corticocortical projections arising from M1, showing labeling specifically in L5A and not in L4.

p.11 – “... IT neurons in the same cell clusters could send output to different target regions ...” – the problem is that the Alexa-based labeling method tends to under-label, so double-projecting cells will be underestimated, perhaps greatly. I.e., limitations with the labeling technique will tend to exaggerate differences in projection patterns.

p.13 – “Remarkably, our results showed that the entire cohort of IT cells (barring the small Car3 cluster), which consist of several subclasses and constitute >70% of all excitatory neurons in the MOp, formed a largely continuous spectrum of cells instead of discrete clusters.” – just to comment on this further: this is a major finding of the paper – but only surprising from an outdated layer-centric view of cell types, not from the modern molecular/projection-class view. This is also the reason why the nomenclature ordering should be IT-L#-subtype#.

p.14 – “complex many-to-many network” – consider acknowledging the Han et al. paper (2018, Nature) showing this for visual cortical projections using MAPseq.

I think I understand (sort of) the rationale for lumping L4/5 together for the molecularly defined excitatory class, but it makes little sense to apply “L4/5” as a label for a cortical layer (e.g. Fig 2A, Fig 4b). There is no “L4/5”. Just label it “L4”.

p.14 – “more compressive projection map” – should be “comprehensive”

Referee #2 (Remarks to the Author):

It appears that members of the BICCN have submitted a set of partially related manuscripts for review on the general topic of cell type in the mammalian brain, including the present manuscript as well as cited references 38, 49, 59-61. These studies use a range of sequencing, imaging, genetic, functional, and other tools to analyze single-cell distributions of transcripts, epigenetic marks, and brain circuitry, which are presented in the form of several inter-related atlases that will be available through publicly-accessible repositories. Overall, this is a massive release of data from multiple leading groups and institutions that will become an invaluable resource for the neuroscience community.

Zhang et al., in the manuscript “Molecular, spatial and projection diversity of neurons in primary motor cortex revealed by in situ single-cell transcriptomics”, use their MERFISH technology to the study of cell types of the mouse primary motor cortex. MERFISH uses sequential rounds of in situ hybridization with carefully barcoded FISH probes to read out, in this instance, hundreds of

transcripts in large areas of thin sections sampled across the mouse brain. The breadth of the study is remarkable in the imaging of ~250 mRNA transcripts in ~300,000 cells that revealed ~100 distinct cell types as well as information about cell-cell connectivity and projections. Their findings support some earlier studies in cell typing using single-cell sequencing of dissociated cells, but go beyond the earlier work by also measuring the distribution of cells within intact tissue and, in some cases, also concurrently measuring projection targets using retrograde tracing with cholera toxin b labeling.

On a technical level, their work is solid. They use ensemble sequencing analyses, together with previous results from the literature (e.g., Allen Institute collaborators' single-cell sequencing cell typing project), to corroborate many of their transcriptomics-based cell type findings. They are using the Allen Inst. Mouse common coordinate framework, have studied multiple mice, have carefully assessed their own reproducibility, etc., which will aid in the utility of this work by the community.

This will be an important reference for many in the community and I encourage its publication. I find that the manuscript is well-written and well-documented and I have embarrassingly few suggestions for improvements. I could imagine a potential criticism that this work is at times descriptive in nature, however, I believe that this is easily countered by the following. First, the strength of their technology enables them to create these massive data sets and there has been excellent work in developing a rigorous data analysis pipeline here. Second, the cell type atlas is one of the most comprehensive catalogs of its sort in the brain to date and will be a highly valuable resource for the community. These dual achievements, in showing the way forward while also providing a valuable resource, will make the work suitable for the broad readership of Nature.

Here are a few very minor points that should be easily addressed by the authors.

1. Spelling would need to be converted from American English to British English in a number of places (neighbourhood, colour, etc.).
2. Figure 4 caption should state "principal-component" rather than "principle-component" as well as "19 principal components".
3. Most figures contain substantial amounts of text (axis labels, legends) that are small and/or faint, making them very hard to read even at 100%. The authors may wish to put some more attention to readability in this regard, although it is inevitable to be difficult for some cases such as when using very large legends.
4. The authors should include a citation for their reference of the BICCN flagship paper on page 9, 5 lines from the bottom (that sentence only cites reference 51, Yamawaki et al. 2014).
5. In the methods section, it may still be valuable to readers to learn the duration of the data acquisition period, although I do realize that this has been covered in some previous MERFISH papers.
6. In the section "Animals", male mice were described. In the section "Stereotaxic injection..." the authors describe both male and female mice. This should be reconciled.

Author Rebuttals to Initial Comments:

Referee #1 (Remarks to the Author):

This study, representing the spatial transcriptomics arm of the BICCN effort to characterize

cortical cells in mouse “MOp”, uses MERFISH to assess gene expression as a function of soma locations across layers. This work has potentially high significance for both motor cortex research and for understanding cortical organization across areas and species. Findings include: (i) evidence against conventional layers playing the primary role as a determinant of cellular identity; (ii) evidence against molecular specification playing a role in the identity of different projectionally defined subtypes of IT neurons projecting to other cortical areas in various patterns (p.12); (iii) an inventory of gene expression of (nearly) all known cell types (importantly including nonneuronal classes) as a function of soma location, mainly in terms of depth from cortex, but also A-P position within MOp; (iv) quantitative information about relative abundances of cell types; (v) further evidence for and characterization of various subtypes of neurons, particularly interneurons, which are relatively under-studied in motor and frontal areas. Thus there are a number of important results. Comments, concerns, and suggestions are as follows.

Response: We thank the reviewer for his/her enthusiasm about our work and for the constructive comments, which we address point-by-point below and in our revised manuscript. The additional analyses and text revisions we did to address these comments have provided additional support for our conclusions and further strengthen our paper.

Spatial analysis: How exactly were the spatial analyses performed? How was soma depth and all the related morphometric parameters (depth of cortex, etc) actually measured? Was the ‘sloping’ of layers (well known in MOp, particularly in the medial part, and clearly evident in the figures) and variation in cortical thickness taken into account and compensated for in the analyses? How were layers defined and identified? More on these points below.

Response: From the MERFISH images, we segmented the cells and determined the centroid coordinates of all cells. For each cell, the soma depth was determined as the shortest distance of its centroid position to the cortical surface, which is marked by the very thin layer of VLMC cells. In other words, we calculated the soma depths of individual cells along the direction perpendicular to the cortical surface. We took into account the variation in cortical thickness from slice to slice and compensated for that in our analyses. To do this, we measured the cortical thickness in each coronal slice, which was defined as the median soma depths of the thin layer of L6b cells in the slice and the soma depth of each cell was normalized by the cortical thickness of the slice. We described how soma depth of cells were calculated in Figure 2 caption, and now added a description of this to the Methods section too (page 55, “Soma depth determination” section). More discussions on the effect of cortical-thickness and layer-thickness variations along the medial-lateral direction will follow later in this response letter.

On how layers were defined and identified: We classified the cells based on their molecular signatures (i.e. gene expression profiles) alone and did not use spatial information in cell classification, hence the main conclusions of this paper do not depend on the precise positions of these layer boundaries. The approximation of the layer boundaries shown in the figures were determined based on the boundaries of the layer-specific excitatory neuronal cell types, in particular the L2/3 IT, L4/5 IT, L5 IT and ET, L6 IT and CT, and L6b cells. We classified these cell types by their expression of previously identified layer-specific marker genes and their correspondence with the sc/snRNA-seq clusters defined in a companion paper (Yao et al, 2020, Ref 38) in this BRAIN Initiative Cell Census Network (BICCN) package. Once these cell types were classified, we determined the approximate layer positions as follows. L1 extends from the cortical surface (i.e. the median soma depth of the VLMC cells) to the top edge of the L2/3 IT neurons

(e.g. the median soma depth of the most superficial L2/3 IT cluster, L2/3 IT 1), L2/3 extends from the top edge of the L2/3 IT neurons to the top edge of the L4/5 IT neurons (e.g. the median soma depth of the L4/5 IT 1 cluster), L4 extends from the top edge of the L4/5 IT neurons to the top edge of the L5 IT and ET neurons (e.g. the median soma depth of the L5 IT 1 cluster), L5 extends from the top edge of the L5 IT and ET neurons to the top edge of the L6 IT and CT neurons (e.g. the median soma depth of the L6 IT 1 cluster), and L6b layer is defined by the distribution of the L6b neurons. We now clarified how layer boundaries were estimated in the Methods section in the revised manuscript (Pages 55-56, "Layer boundary assessment" section). We also note that some of the L4/5 IT clusters, such as L4/5 IT 5, may belong to L5 (as discussed later in this response letter) and hence we described an uncertainty range of the L4/L5 boundaries due to this cluster in the same section (Pages 55-56, "Layer boundary assessment" section).

On the sloping of the layers: Even though we calculated the soma depth along the direction that is perpendicular to the cortical surface, there is still a modest variation in layer thicknesses along the medial-lateral (ML) direction. If this is what the reviewer meant by "sloping" of layers, this "sloping" was not taken into account in our original analyses. In light of this reviewer comment, to assess how the layer-thickness variation might affect our results, we divided the MOp region into six segments at different ML positions, so that the layer thickness variation within each segment is negligible. We then determined the cortical depth distributions of individual cell clusters in each of the six segments (cortical depth of each cell was still determined along the direction perpendicular to the cortical surface). For illustrative purposes, we showed the cortical-depth distributions for the most medial segment (blue, ML position = 1) and the most lateral segment (orange, ML position = 6) in Extended Data Figure 3c (upper panel). While we could observe some minor shifts in these distributions between segments at different ML positions, overall the distributions look similar at all six different ML positions for most clusters. Only a small number of clusters (such as L4/5 IT SSp1/2, L5 ET 4, and L6 CT 8) exhibited relatively large differences at different ML positions, but this is not due to layer-thickness variations, but due to the region-dependent presence of these clusters. For example, L4/5 IT SSp1/2 and L5 ET 4 is present primarily on the lateral side and rarely on the medial side of the MOp, and L6 CT 8 cluster is present only on the medial side but not the lateral side of the MOp. We have also performed similar analysis for the MOp upper limb (MOp-ul) region and observed similar results, i.e. we observed only relatively minor shifts in the cortical depth distributions between segments at different ML positions for most clusters, except for a few exceptions that showed larger differences due to region-dependent presence of these clusters (Extended Data Figure 3c, lower panel). We note that in Extended Data Figure 3c, we only showed the distributions of the glutamatergic clusters. Because of the relatively low abundance of the GABAergic neurons, the numbers of cells in individual GABAergic clusters in each ML segment are relatively small, which makes comparison of their distributions in different ML segments statistically less sound, and hence these clusters are not shown here.

Sloping layers: Did the sloping of the layers in MOp contribute to blurring of what are actually sharper laminar distributions? E.g. in Fig 3, the distributions of the labeled cells are very different on the far left vs the middle and right of the section; were the distributions in the plot on the right artifactually broadened by pooling these together? This consideration is a general one, applying to all the depth-of-soma analyses. In the extreme, does the "continuous spectrum" of the IT gene expression to some extent reflect artifactual 'smearing' of laminar distributions?

Response: First, we note that we calculated the soma depth along the direction that is perpendicular to the cortical surface, and with this definition, the layer-thickness variations along

the medial-lateral (ML) direction are relatively small. However, such variations were indeed present and could cause some broadening of the cortical depth distributions of individual cell clusters. We thus added Extended Data Figure 3c to show the differences in cortical depth distributions at different ML positions for all glutamatergic neuronal clusters in the MOp. As described in our response to the comment above, we observed relatively minor differences in the cortical depth distributions between segments at different ML positions for most clusters, with only a few exceptions due to the region-dependent presence of these clusters. Since the cortical depth distributions of each glutamatergic neuronal cluster at different ML positions are shown in Extended Data Figure 3c, we did not separately provide these distributions at different ML positions for the clusters in Figure 3. However, in the results section related to Figure 3 (i.e. the “Diversity of L5 ET, L5/6 NP, L6 CT and L6b neurons” section), we pointed out all cluster that exhibited region-dependent presence, such as L5 ET 4, L6 CT 8, etc.

Regarding whether the “continuous spectrum” of the IT cells could be caused by this “sloping” effect, we note that we draw the conclusion on the IT cell gradient based on two observations: (i) in the gene expression space, the 19 IT clusters form a largely continuous cloud, except for L6 IT Car3 which forms an isolated cluster (Figure 4a and Extended Data Figure 8a); (ii) along the cortical depth position, individual IT clusters partially overlapped in space with adjacent clusters (Figure 4b and Extended Data Figure 8b). For the first observation, since we did not use any spatial information in cluster classification, the layer-thickness variation along the ML direction would not have any effect on this observation. For the second observation, we performed additional analyses to check whether the spatial overlap between clusters could be caused by the layer-thickness variations along the ML direction. For that, we divided the MOp into six segments at different ML positions, as described above, and considered IT neurons separately in each segment, within which layer-thickness variation is negligible. We still observed substantial spatial overlap between neighboring IT clusters in each of the six ML segments (see Extended Data Figure 8c). Furthermore, we performed similar analysis for the MOp upper limb (MOp-ul) region and observed that the cortical depth distributions of neighboring IT clusters still show significant overlap within each of the six ML segments in the MOp-ul (Extended Data Figure 8d). Moreover, in the MERFISH-derived cell type maps of individual coronal slices, spatial overlap can also be clearly observed between neighboring subclasses of cells and between neighboring clusters within subclasses, as shown in Figure 4c. Hence, our observations are unlikely due to broadening (or ‘smearing’) of laminar distributions caused by sloping (i.e. thickness variations) of layers.

“MOp”: What is this region, exactly? I understand that it’s the Allen Atlas’s parcellation, but what’s it based on? Why is the border between MOp and MOs drawn where it is? It doesn’t seem to be based on any clear functional or even anatomical considerations. The article could potentially be strengthened by focusing (mainly, or at least in a sub-analysis) on the forelimb region within MOp. Currently, inclusion of the most anterior, lateral, and medial regions of “MOp”, which don’t seem to be well-defined as “primary” motor areas, makes it hard to interpret the findings in terms of their significance for “primary” motor cortex. The authors may wish to point out that “MOp”, as defined here, is not necessarily equivalent to “primary motor cortex”, at least not as it’s most commonly defined in the literature (e.g. based on ICMS thresholds, presence of corticospinal neurons, functional imaging/electrophysiology, etc.).

Response: We thank the reviewer for this suggestion. We first would like to clarify that to ensure that the entire MOp is covered, our imaged region includes both MOp and some adjacent areas. Hence some of our images might be confusing since they clearly include areas adjacent to the

MOp. Our original analyses were done on the entire imaged region, but we pointed out those clusters that are primarily present outside MOp or exhibit significant differences in different parts of the MOp. In light of this comment from the reviewer, we registered our MERFISH images to the Allen Mouse Brain Common Coordinate Framework version 3 (CCF v3, Ref 40) and added additional analyses that focus on the MOp region per se. In this Allen CCF v3, the brain regions are parcellated using multimodal reference datasets including histology stains, immunohistochemistry, transgene expression, and connectivity experiments, which are registered to the average template, with in situ hybridization and other published results also consulted. We used this Allen CCF v3 to define the MOp boundary as it is collectively decided by the BICCN for this consortium effort of MOp studies, and other modalities of measurements presented in other companion papers and the flagship paper of this BICCN package used the MOp definition in this CCF as well. Registration of our MERFISH images to the CCF allowed us to perform quantitative analyses not only for the entire imaged region but also for the MOp. In addition to presenting analysis results for the entire imaged region, in the revised manuscript, we now marked the MOp region in our MERFISH images in the figures to avoid confusion and also presented results for many of these analyses for the MOp region per se in the Extended Data Figures 2c, 3, 4b, 8 and 9. We also explicitly pointed out in our manuscript that the MOp region studied in this work is referring to the MOp defined in the Allen CCF v3. All of the 95 cell clusters described in Fig. 1b were also found in the MOp. Although a few clusters are enriched outside of the MOp, such as L4/5 IT SSp 1/2, L5 ET 4, and L6 CT 8, they were still found to be present in the MOp. The cortical depth distributions of the cell clusters derived from cells within the MOp appear similar to the cortical depth distributions derived from the entire imaged region (Compare Extended Data Figure 3a with Figure 2c,d).

We also took the reviewer's suggestion to perform some analyses on the upper limb (forelimb) region of the MOp (MOp-ul). Since there is no reference map of the MOp-ul in the CCF v3 to help us define precise boundaries of the MOp-ul, we selected the region between Bregma 0 and +1.0 within the MOp as an approximation for the MOp-ul, and this selection is based on results from previous literatures (Refs 67-69) and a companion BICCN paper (Ref 53). This region is considered the primary part of the MOp-ul because it contains the densest pyramidal neurons that project directly to the intermediate horn and ventral horn of the cervical spinal cord and, in the meantime, shows minimal projections to the lower limb (Ref 53). This region also corresponds to the caudal forelimb domain (CFA) defined based on intracortical microstimulation (ICMS) evidence in the mouse brain (Ref 67). As this is an approximation for the MOp-ul, we performed only a limited set of analyses on this region and presented these analysis results in revised manuscript (i.e. the cortical depth distributions of the neuronal clusters in Extended Data Figures 3b, c and 8d). 93 out of the 95 cell clusters that we observed in the MOp were also found in this MOp-ul region. The cortical depth distributions of the clusters within this region also appear similar to the cortical depth distributions derived from cells in the entire MOp (Compare Extended Data Figure 3b with Extended Data Figure 3a). Our study of MOp projections to MOs, SSp and TEa by integration of MERFISH with retrograde labeling was also performed in this MOp-ul region (Figure 5).

“L5”: What exactly is this, and how does it relate to the much more precise designation of a distinct L5A and L5B, which is often used in studies of mouse motor, frontal, and somatosensory cortex? I understand that these layers are indistinct in some regions of the mouse cortex where

layers appear merged (e.g. cingulate cortex, occipital areas), but in most areas and species there is a clear distinction between L5A and L5B, and this is certainly the case for mouse M1. Up to the authors, but I would recommend the following laminar definitions: L1: easily defined (e.g. absence of PyN); L2/3: everything between L1 and L4; L4: RORB labeling; L5A: *Etv1* and absence of PT neurons; L5B: presence of PT neurons; L6A: presence of CT neurons; L6B: as you have done. For example, in Extended Data Figure 6 the full laminar distribution of the PT neurons defines L5B (which is much broader in MOp than SSp).

Response: We defined L5 in this manuscript by the locations of L5 ET and L5 IT neurons. As mentioned earlier, we classified cells based on their gene expression profiles, and did not use spatial information as input for cell clustering. We then provided an approximate layer boundary assessment based on the spatial distributions of L2/3 IT, L4/5 IT, L5 IT and ET, L6 IT and CT, and L6b cells, as described earlier in this response letter (now also clarified in the Methods section, Pages 55-56, "Layer boundary assessment" section). We did not find *Rorb* to be a very specific L4 marker because *Rorb* labels both L4/5 IT and some L5 IT clusters, as observed both in our MERFISH data (Figure 1b) and the sc/snRNA-seq data in the companion paper (Ref. 38).

Since we did not include the L5a marker *Etv1* in the panels of genes imaged by MERFISH, precise separation of L5a and L5b is challenging for us. Instead of providing a precise delineation of L5a and L5b, we examined which of the L4/5 IT and L5 IT clusters might belong to L5a by comparing their distributions with the L4 marker *Rspo1*, L5 marker *Fezf2* and the distributions of L5 ET (PT) cells. Based on these results, we found that L5 IT 1 may partially reside in L5a because of its low degree of overlap with *Rspo1* and strong overlap with *Fezf2*, and its significantly lower overlap with L5 ET cells than other L5 IT clusters. In addition, L4/5 IT 5 likely resides in, or partially resides in, L5a because of its overlap with *Rspo1* is substantially lower than those of the other L4/5 IT clusters (L4/5 IT 1-4), its overlap with *Fezf2* is substantially higher than those of L4/5 IT 1-4 clusters and comparable to those of several other L5 IT clusters, and its overlap with L5 ET cells is low. These results are now presented in Extended Data Figure 10 and described in Methods (Pages 55-56, "Layer boundary assessment" section). Because some L4/5 IT cells, such as the L4/5 IT 5 cluster, potentially reside in L5, there is also an uncertainty in the location of the L4 and L5 layer boundary, which is also now clarified in Figure 2 and in the Methods section (Pages 55-56, "Layer boundary assessment" section).

Nomenclature: Why introduce the new and much less precise term "ET" to replace the well-established "PT" nomenclature? To be sure, "PT" is an imperfect term – as are "neocortex", "isocortex", and many other everyday terms in cortical neuroanatomy. Yet "PT" is widely used and hardly problematic especially in motor cortex research, where it's been the preferred term for over half a century. If you really want to replace it, there's the equivalent and more technically precise (but more of a mouthful) term "subcerebral projection neurons". The problems with "ET" are multiple, and hardly worth going into, but in a nutshell: it implies that these neurons don't form IT projections (they do), and that they alone form ET projections (not so, as CT neurons do, too), and it tries (erroneously in this case) to define something not by what it is but by what it's not ("not-IT").

Response: In this manuscript, we followed the nomenclature style that is collectively decided by the BICCN so that our cluster nomenclature is consistent with that used in the flagship paper and other companion papers in this BICCN package. The rationale for using the term L5 ET is presented in the BICCN flagship paper. Please see the "Nomenclature of the L5 ET subclass of

glutamatergic neurons” section in the Supplementary Notes of the flagship paper (Ref 37: bioRxiv, <https://doi.org/10.1101/2020.10.19.343129>). We have conveyed this reviewer comment to the BICCN flagship paper working group. If after the review process of the flagship paper, the BICCN decides to change the name of this subclass of neurons, we will also change it accordingly in our paper.

Minor:

Taxonomic naming conventions: order should go from major to minor. E.g. IT-L2/3-subtype, not L2/3-IT-subtype1, etc. For PT and CT, no need to indicate layer, since they are by definition in one particular layer, but can indicate subtype/sublayer as appropriate (e.g. PT-subtype1, PT-subtype2, etc).

Response: We do see the reviewer’s point, and agree with the reviewer’s assessment on which property is more major in cell-type classification. By placing the layer information before the projection information in the cluster names, we do not mean to imply that the layer distinction is more important than the projection distinction. We followed the nomenclature style that is collectively decided by the BICCN so that the cluster nomenclature in our paper is consistent with that used in the flagship paper and other companion papers in this BICCN package. We have conveyed this reviewer comment to the BICCN flagship paper working group. If after the review process of the flagship paper, the BICCN decides to change the nomenclature style of the clusters, we will also change accordingly in our paper.

Corticospinal neurons: The archetypal cell type of primary motor cortex, yet never mentioned (in what is intended as a “comprehensive” analysis of MOp cell types). Consider at least a brief mention, e.g. with citation to the closely related Bakken et al. paper (ref. 59). Could point out that there’s no unique molecular marker for corticospinal neurons.

Response: As suggested by the reviewer, we added a mentioning of the corticospinal neurons in the revised manuscript (Page 9, 1st paragraph): “The corticospinal projection neurons, homologous to the Betz cells in primates (Ref 51), are likely also contained in the L5 ET subclass (Ref 52). However, because of the lack of unique molecular marker for these neurons, it is unclear which of the L5 ET clusters identified here are corticospinal neurons. Spinal-projecting neurons are shown to be dorsal to the medulla-projecting neurons in layer 5 of the MOp in a BICCN companion paper (Ref 53), it is thus possible that corticospinal neurons are more enriched in the L5 ET 1-3 clusters.” We cited the related Bakken et al. paper (currently Ref. 51) here, as well as a couple of other references here.

p.3, “Single-cell gene expression...” section, first paragraph – this is basically a description of the recipe for generating the gene panel. But what’s the rationale for the particular blend of the three subsets of genes? Surely much thought went into this, so it would be nice to know the thinking behind it. Also, citations be provided for the “prior knowledge”?

Response: We selected three subsets of genes: (I) canonical marker genes for major neuronal and non-neuronal cell types in the cortex selected based on prior knowledge, (II) a set of genes (50 for glutamatergic and 50 for GABAergic neuronal clusters) which contained the highest mutual

information among the clusters identified by sc/snRNA-seq (referred to as MI genes below, MI for mutual information), (III) cluster marker genes selected based on pair-wise differential gene expression analysis on the neuronal clusters identified by sc/snRNA-seq (referred to as DE genes below, DE for differential expression);

Subset I contains the major marker genes that we can find in the literature and we now added references for this subset as suggested by the reviewer. For subsets II and III based on sc/snRNA-seq results, we note that subset II (the MI genes) tend to be genes that are differentially expressed between groups of cell clusters, whereas subset III (DE genes) are differentially expressed between individual pairs of clusters. These two sets could have complementary power and, when combined, could give better cluster identification results in our experience. The number of genes in these two subsets were empirically chosen. For MI genes, we selected the top 50 genes with highest mutual information for excitatory neuronal clusters and the top 50 genes for inhibitory neuronal clusters, and due to overlap between the two groups, this approach generated a total of 91 genes. For the DE genes, 168 genes were selected by including at least 2 DE genes for each cluster pair in both directions.

These additional descriptions of rationales for gene selection are now included in the Methods section of the revised manuscript (See Methods, “Gene selection for MERFISH” section, pages 44-45). We also added a supplementary table (Supplementary Table 1) to list all selected genes and which one(s) of the three subsets each gene belongs to.

p.5 – cell class abundances are given as percentages; what are the 95% CI?

Response: As suggested by the reviewer, we added 95% confidence intervals to these percentages and described them in the caption of Extended Data Figure 2b,c.

Sometimes unclear use of the term “layer” – e.g. p.6, does “layers” refer to one of the supposed “6 layers”, or to “strata” or perhaps “sublayers”?

Response: In the revised manuscript, we tried to consistently use the term “layers” only for referring to the canonical cortical layers, i.e. the supposed “6 layers”.

p.7 – the “high level of local cellular heterogeneity” – is this reflective of MOp in particular? Or similar to the “salt-and-pepper” organization of mouse V1? Is it a mouse thing in general?

Response: This heterogeneity is probably not a MOp specific feature. For example, in our previous MERFISH study of the mouse hypothalamic preoptic region (Ref 36) we also found high degree of spatial intermixing of cell clusters. This is not necessarily the same thing as the “salt and pepper” pattern in mouse V1. The “salt and pepper” organization in mouse V1 describes the spatial mixing of neurons that respond to visual stimuli of different orientations, but it is not clear whether these neurons responding to visual stimuli of different orientations are different cell types as defined by molecular signatures from single-cell expression files.

p.7 – “L5 ET 4 mostly resided outside of MOp” – where, exactly?

Response: L5 ET 4 is mostly found at the lateral/ventral side of MOp. This cluster was absent from the MOp for most of the imaged slices, and only begun to extend into the MOp on the lateral and

ventral side from the agranular insular area (Ald) in the anterior slices. We now provide the overlay of MERFISH images on the Allen CCF v3 in Extended Data Figure 6c, and revised the manuscript text to make this point clearer (Page 8, last paragraph).

p.8 – “unique medulla-projecting L5 ET 5 cell type was mainly present in MOp but rarely in the adjacentSSp region” – this is surprising; suggests that there different types of medulla-projecting PT cells in MOp and SSp? Cf. Oberlaender papers.

Response: We thank the reviewer for pointing this out. Although we did observe that the L5 ET 5 cell cluster is less abundant in SSp, saying that it is rarely present in the SSp is not accurate. As illustrated in Extended Data Figure 6d, L5 ET 5 cells are less abundant in the SSp, and also appear to show more spatial overlap with L5 ET 1-3 clusters in the SSp than in the MOp. Now that we registered our images to the Allen CCF v3, we separately quantified the cells that fell within the MOp and SSp, and calculated the fraction of cells in MOp that belong to the L5 ET 5 cluster and the fraction of cells in SSp that belong to the L5 ET 5 cluster. We found that the proportion of L5 ET 5 cells in the MOp was ~ 3 times as many as that in the SSp. These additional analysis results are shown in Extended Data Figure 6d. Our result is corroborated by a BICCN companion paper recently posted in bioRxiv (Ref. 50), which shows that medulla-projecting ET neurons are substantially less abundant in SSp than in MOp and MOs.

p.9 – “it is also possible to impute marker genes ... which would ... provide a more complete marker gene sets for the MERFISH clusters” – and would give better nomenclature; i.e., gene names instead of subtype numbering (1, 2, etc.), which is hard to remember, confusing, and will likely need revision anyways. Since this is intended as the definitive study (or set of studies), isn't now the time to identify the marker genes and use them for naming clusters/subtypes?

Response: While it would indeed be helpful to replace the subtype numbering in the cluster names with marker gene names, it is however challenging based on our current data. Sophisticated computational analysis is required to impute expression profiles of genes not imaged in the MERFISH clusters by integration of MERFISH and scRNAseq results, which is outside the scope of this manuscript. This is carried out by computational biology labs in the BICCN, and these results will be published separately.

In general, providing more meaningful cell cluster names beyond the subclass level is a highly non-trivial task and requires synthesis of results from different modality of measurements. The BICCN flagship paper (Ref 37: bioRxiv, <https://doi.org/10.1101/2020.10.19.343129>) tries to do this to some extent by providing marker genes and anatomical/projectional annotations for some (but not all) of the clusters defined by sc/snRNAseq. The correspondence between MERFISH and scRNAseq clusters (albeit not always one-to-one) could be used to provide additional information for some of the MERFISH clusters, as well as spatial information for some of the scRNAseq clusters, as described in the flagship paper.

Future studies that fully integrate the MERFISH and scRNAseq data will hopefully help provide more comprehensive nomenclature for the cell clusters identified by MERFISH.

p.9 – IT section: is soma depth used for the clustering analysis? and/or layer? or is the clustering done agnostic to soma depth, and the clusters turn out to have laminar distributions? This should

be clarified at the outset. Also, how was classification done? By stipulating the number of layers first? How would it differ if layer 4, 5A, 5B were considered separate layers?

Response: Soma depth and layer information were not used in our cell clustering analysis. Clustering analysis of cells was performed purely based on their gene expression profiles, and the clusters just turned out to have laminar distributions. We now clarified this point in the revised manuscript (page 4, last paragraph and Page 5, 1st paragraph). The naming of subclasses was done after clustering, based on their expression of layer-specific marker genes as well as their correspondence to the cell subclasses and clusters derived from sc/snRNAseq data. Because the clustering of cells was performed without using soma depth or layering information of the cells, adopting a different layer delineation (such as L5a and L5b) would not affect the clustering results. For naming the clusters, we followed the nomenclature style set by the BICCN, which does not distinguish L5a and L5b in cluster names, hence this distinction does not appear in the cluster nomenclature in this package of BICCN flagship and companion papers, including ours.

p.11 – choice of “MOs” and “SSp” makes sense but “TEa” seems like an odd choice, compared to for example S2, which has relatively well-characterized interconnections with M1 and S1. Moreover, how cleanly can TEa be injected using a vertical (dorsal) approach as seems to have been used here? And, wouldn't such an approach result in labeling at only one laminar level, which could affect the results if axons ramifying in other layers went unlabeled?

Response: We note that this part of the paper, i.e. the integration of MERFISH imaging with retrograde labeling, is designed to be a proof-of-principle study, instead of a comprehensive projection mapping. We chose temporal association area (TEa) as one of the three target regions because previous retrograde tracing experiments showed that the retrograde labeling pattern in the MOp with TEa injection displayed an interesting laminar distribution in multiple, separate layers (Ref. 9), so we were interested in finding out what these spatially distinct TEa-projecting neurons are.

Indeed, as the reviewer alluded to, dorsal injection to the TEa typically spreads into neighboring ectorhinal area (ECT) and perirhinal area (PERI), too. Previously we have observed that both TEa and ECT are targeted by MOp neurons (Ref. 9), and that injections that are more in ECT or more in TEa yielded the same pattern of cells in ipsilateral MOp. Single neuron reconstruction studies also found that axons that target TEa frequently collateralize to ECT and PERI. So these regions are often referenced together in the literature as a "complex" (e.g. Refs 9, 53, and 59). Without using more sophisticated anatomical methods, these structures are difficult to separate. Nonetheless, we should indeed make it clear that ECT and PERI regions are also labeled in the TEa injections and have revised our manuscript to clarify this point (page 12, 3rd paragraph). We thank the reviewer for pointing this out.

With regard to whether the injections labeled only one laminar level, we note that for the TEa injection, all layers were labeled by CTb. MOs and SSp injections covered most of the layers, but the CTb stain appeared relatively weak in L1 and part of L6 in the MOs and SSp, and hence projection to L1 and L6 of MOs and SSp may be under-represented. We also clarified this in the revised manuscript (See Methods, “Stereotaxic injection of retrograde tracers” section, page 58).

p.11 – “... enriched in upper layers L2/3 and L4/5; ... enrichment in upper L2/3, deep L4/5 and L6” – this sentence exemplifies the problems with the lack of recognition of L5A – previous studies

have described labeling of corticocortical projections arising from M1, showing labeling specifically in L5A and not in L4.

Response: As suggested by the reviewer, the deep L4/5 region could indeed be L5A. In our response to an earlier comment from this reviewer, we noted that L4/5 IT 5 and L5 IT 1 could reside, or partially reside, in L5A. The most abundant clusters that projected to TEa, as observed in our study, are L4/5 IT 5 and L5 IT 1 (along with L6 IT 3 and L2/3 IT 1-3), which is consistent with previously results showing retrograde labeling in L5A.

We revised the manuscript to point out that L4/5 IT 5 and L5 IT 1 may reside, or partially reside, in L5A (See Methods, Pages 55-56, "Layer boundary assessment" section).

p.11 – "... even neighboring IT neurons in the same cell clusters could send output to different target regions ..." – the problem is that the Alexa-based labeling method tends to under-label, so double-projecting cells will be underestimated, perhaps greatly. I.e., limitations with the labeling technique will tend to exaggerate differences in projection patterns.

Response: Indeed, as pointed out by the reviewer, this dye labeled CTb-based retrograde labeling method could under-label projection neurons, and thus double-projecting neurons could be underestimated. Hence, our observation that "even neighboring IT neurons in the same cell clusters could send output to different target regions" could be in part due to double-projecting neurons that appeared as single-projecting because of under-labeling. We thus deleted this sentence from the manuscript. In fact, since this is not a major finding in our paper, removing this sentence actually improves the readability of this part of the paper. To acknowledge this limitation in the retrograde labeling method, we now explicitly point out that dye labeled CTb-based retrograde labeling could under-label projection neurons, and thus double-projecting neurons could be under-represented by this labeling approach in the revised manuscript (see the Methods, "Stereotaxic injection of retrogradetracers" section, page 58). We thank the reviewer for pointing this out.

It is however worth noting that our conclusion that each cell cluster can project to multiple different regions is still accurate. In fact, our major conclusions in this section of the paper, i.e. individual cell clusters can target multiple regions, individual targeted regions can receive input from multiple cell clusters, and some molecularly and spatially similar clusters show highly distinct projection properties are all still valid. Moreover, even though we may not be observing all of projecting neurons due to under-labeling by CTb, our quantification of the composition of cell clusters projecting to each target region and projection patterns of each cell cluster are still most likely accurate, because we imaged and quantified a very large number of cells (~190,000 cells from 2 animal replicates) for these projection measurements.

p.13 – "Remarkably, our results showed that the entire cohort of IT cells (barring the small Car3 cluster), which consist of several subclasses and constitute >70% of all excitatory neurons in the MOp, formed a largely continuous spectrum of cells instead of discrete clusters." – just to comment on this further: this is a major finding of the paper – but only surprising from an outdated layer-centric view of cell types, not from the modern molecular/projection-class view. This is also the reason why the nomenclature ordering should be IT-L#-subtype#.

Response: As explained earlier, with the cluster nomenclature style used in our manuscript, which

place the layer information before the projection information, we do not mean to imply that the layer distinction is more important than the projection distinction in cell-type classification. We agree with the reviewer that the projection distinction is more basic than the layer distinction here. We followed the nomenclature style that is collectively decided by the BICCN so that the cluster nomenclature in our paper is consistent with that used in the flagship paper and other companion papers in this BICCN package. We have conveyed this reviewer comment about cluster nomenclature to the BICCN flagship paper working group. If after the review process of the flagship paper, the BICCN decides to change the nomenclature style of the clusters, we will also change accordingly in our paper.

p.14 – “complex many-to-many network” – consider acknowledging the Han et al. paper (2018, Nature) showing this for visual cortical projections using MAPseq.

Response: We added this reference (Ref 63) as suggested.

I think I understand (sort of) the rationale for lumping L4/5 together for the molecularly defined excitatory class, but it makes little sense to apply “L4/5” as a label for a cortical layer (e.g. Fig 2A, Fig 4b). There is no “L4/5”. Just label it “L4”.

Response: As suggested by the reviewer, we changed the layer label from L4/5 to L4. As mentioned earlier, since the L4/5 IT 5 cluster could be residing, or partially residing, in L5 (L5a), we also marked the uncertainty of the boundary position between L4 and L5 due to the L4/5 IT 5 cluster in the figures.

p.14 – “more compressive projection map” – should be “comprehensive”

Response: We thank the reviewer for spotting this typo and corrected it in the revised manuscript.

Referee #2 (Remarks to the Author):

It appears that members of the BICCN have submitted a set of partially related manuscripts for review on the general topic of cell type in the mammalian brain, including the present manuscript as well as cited references 38, 49, 59-61. These studies use a range of sequencing, imaging, genetic, functional, and other tools to analyze single-cell distributions of transcripts, epigenetic marks, and brain circuitry, which are presented in the form of several inter-related atlases that will be available through publicly-accessible repositories. Overall, this is a massive release of data from multiple leading groups and institutions that will become an invaluable resource for the neuroscience community.

Zhang et al., in the manuscript “Molecular, spatial and projection diversity of neurons in primary motor cortex revealed by in situ single-cell transcriptomics”, use their MERFISH technology to study cell types of the mouse primary motor cortex. MERFISH uses sequential rounds of in situ hybridization with carefully barcoded FISH probes to read out, in this instance, hundreds of transcripts in large areas of thin sections sampled across the mouse brain. The breadth of the study is remarkable in the imaging of ~250 mRNA transcripts in ~300,000 cells that revealed ~100 distinct cell types as well as information about cell-cell connectivity and projections. Their findings support some earlier studies in cell typing using single-cell sequencing of dissociated cells, but go

beyond the earlier work by also measuring the distribution of cells within intact tissue and, in some cases, also concurrently measuring projection targets using retrograde tracing with cholera toxin b labeling.

On a technical level, their work is solid. They use ensemble sequencing analyses, together with previous results from the literature (e.g., Allen Institute collaborators' single-cell sequencing cell typing project), to corroborate many of their transcriptomics-based cell type findings. They are using the Allen Inst.

Mouse common coordinate framework, have studied multiple mice, have carefully assessed their own reproducibility, etc., which will aid in the utility of this work by the community.

This will be an important reference for many in the community and I encourage its publication. I find that the manuscript is well-written and well-documented and I have embarrassingly few suggestions for improvements. I could imagine a potential criticism that this work is at times descriptive in nature, however, I believe that this is easily countered by the following. First, the strength of their technology enables them to create these massive data sets and there has been excellent work in developing a rigorous data analysis pipeline here. Second, the cell type atlas is one of the most comprehensive catalogs of its sort in the brain to date and will be a highly valuable resource for the community. These dual achievements, in showing the way forward while also providing a valuable resource, will make the work suitable for the broad readership of Nature.

Response: We thank the reviewer for his/her enthusiasm about our work and for the constructive suggestions, which we address point-by-point below and in our revised manuscript. The text revisions we did to address these comments have improved the presentation of our results.

Here are a few very minor points that should be easily addressed by the authors.

1. Spelling would need to be converted from American English to British English in a number of places (neighbourhood, colour, etc.).

Response: We made the changes as suggested.

2. Figure 4 caption should state "principal-component" rather than "principle-component" as well as "19 principal components".

Response: We thank the reviewer for spotting these typos and corrected it in the revised manuscript.

3. Most figures contain substantial amounts of text (axis labels, legends) that are small and/or faint, making them very hard to read even at 100%. The authors may wish to put some more attention to readability in this regard, although it is inevitable to be difficult for some cases such as when using very large legends.

Response: we have rearranged some figure panels and increased the sizes of the figure legends and axis labels in the figures to improve readability.

4. The authors should include a citation for their reference of the BICCN flagship paper on page 9, 5 lines from the bottom (that sentence only cites reference 51, Yamawaki et al. 2014).

Response: This flagship paper was not ready to be cited when we submitted our paper. Now that the flagship paper is out on BioRxiv, we added the citation information (Ref 37).

5. In the methods section, it may still be valuable to readers to learn the duration of the data acquisition period, although I do realize that this has been covered in some previous MERFISH papers.

Response: we have added this information to the Methods section (Page 50, 1st paragraph).

6. In the section "Animals", male mice were described. In the section "Stereotaxic injection..." the authors describe both male and female mice. This should be reconciled.

Response: The mice used in the stereotaxic injection for the retrograde tracing experiments were also males. We apologize for this typing error and have corrected it in the Methods section ("Stereotaxic injection of retrograde tracers" section, page 58).

Reviewer Reports on the First Revision:

Referees' comments:

Referee #1 (Remarks to the Author):

The revisions have improved the study in some though not all respects, but also expose a remarkable and serious problem: the borders of MOp and particularly MOp/SSp, have been drawn in a clearly erroneous manner. This is remarkable because this border is not at all hard to draw correctly, and serious because it constitutes a major flaw in how cells were sampled.

The MOp/SSp border, like most cortical area borders, is roughly perpendicular to the cortical surfaces and layers. Certainly there can be deviations from this where the cortex is curved, but on the dorsolateral surface of the cortex where most of the MOp/SSp border is found this border is close to perpendicular. This is well known and easily seen in the apical dendrites of pyramidal neurons, a reflection of the radial development of the cortex. An example of how to draw the MOp/SSp reasonably correctly is the current Allen Reference Atlas (v2, 2011), e.g. including images 43-52/132.

The current paper bizarrely draws the MOp/SSp border diagonally at an angle of about 45 deg to the cortical surface, as shown in the tiny images from the Allen Brain Atlas in Fig 1a, Fig 1b, and indicated by the grayed-out zones in Fig 2a,b. Because of this oddly angled border, the sampling includes SSp neurons, and increasingly so as a function of cortical depth; i.e., the "L6" population includes many cells that are actually located in SSp, and presumably excludes some MOp L6 cells by incorrectly assigning them to MOs. The more superficial cells in MOp may conversely have been under-sampled. Fig 3 also shows how tilted the borders are, as does Ext Data Fig 2a (and Ext Data Fig 6c, Ext Data Fig 7c,d). The Ext Data Fig 6c shows how extreme and even absurd the problem is – the way the borders are defined here, MOp is defined as an oddly shaped piece of cortex that

completely undercuts both SSp and even GU cortex, at an angle almost parallel to the layers. This is literally borderline nonsense; areal borders don't run with the layers, they cut across them.

Because of these patently incorrect border definitions, the "MOp" sampling is clearly inaccurate as it is contaminated with cells (particularly in deeper layers) not actually located within MOp, and probably excludes bona fide MOp cells (particularly in superficial layers). As this would seem to pertain to the datasets in their entirety, I see no alternative but to recommend a complete re-analysis based on properly drawn MOp boundaries, so that the sample contains all MOp neurons and only MOp neurons, without contamination from SSp or MOs.

Regarding the response in the rebuttal, "... We used this Allen CCF v3 to define the MOp boundary ..." – nevertheless areal borders need to be drawn correctly.

Addendum: having just now looked at the Munoz-Casteneda BICCN paper (on bioRxiv, although missing some key supplemental figures), it seems that in at least some if not most of the figures they correctly drew the MOp/SSp border, i.e., roughly perpendicular across the layers (Fig 2f, Fig 3b, among others). They also show in Fig 2c a comparison of three ways of drawing the border, which includes the border based on CCFv3, which one can see is quite different from the "expert manual annotation" version that seems to get it right. (An "algorithmically detected border" looks just as wrong as the CCFv3 based border.)

Referee #2 (Remarks to the Author):

The authors have sufficiently addressed my feedback and I recommend acceptance for publication.

Author Rebuttals to First Revision:

Reviewer 1 comments:

The revisions have improved the study in some though not all respects, but also expose a remarkable and serious problem: the borders of MOp and particularly MOp/SSp, have been drawn in a clearly erroneous manner. This is remarkable because this border is not at all hard to draw correctly, and serious because it constitutes a major flaw in how cells were sampled.

The MOp/SSp border, like most cortical area borders, is roughly perpendicular to the cortical surfaces and layers. Certainly there can be deviations from this where the cortex is curved, but on the dorsolateral surface of the cortex where most of the MOp/SSp border is found this border is close to perpendicular. This is well known and easily seen in the apical dendrites of pyramidal neurons, a reflection of the radial development of the cortex. An example of how to draw the MOp/SSp reasonably correctly is the current Allen Reference Atlas (v2, 2011), e.g. including images 43-52/132.

The current paper bizarrely draws the MOp/SSp border diagonally at an angle of about 45 deg to the cortical surface, as shown in the tiny images from the Allen Brain Atlas in Fig 1a, Fig 1b, and indicated by the grayed-out zones in Fig 2a,b. Because of this oddly angled border, the sampling includes SSp neurons, and increasingly so as a function of cortical depth; i.e., the “L6” population includes many cells that are actually located in SSp, and presumably excludes some MOp L6 cells by incorrectly assigning them to MOs. The more superficial cells in MOp may conversely have been under-sampled. Fig 3 also shows how tilted the borders are, as does Ext Data Fig 2a (and Ext Data Fig 6c, Ext Data Fig 7c,d). The Ext Data Fig 6c shows how extreme and even absurd the problem is – the way the borders are defined here, MOp is defined as an oddly shaped piece of cortex that completely undercuts both SSp and even GU cortex, at an angle almost parallel to the layers. This is literally borderline nonsense; areal borders don't run with the layers, they cut across them.

Because of these patently incorrect border definitions, the “MOp” sampling is clearly inaccurate as it is contaminated with cells (particularly in deeper layers) not actually located within MOp, and probably excludes bona fide MOp cells (particularly in superficial layers). As this would seem to pertain to the datasets in their entirety, I see no alternative but to recommend a complete re-analysis based on properly drawn MOp boundaries, so that the sample contains all MOp neurons and only MOp neurons, without contamination from SSp or MOs.

Regarding the response in the rebuttal, “... We used this Allen CCF v3 to define the MOp boundary ...” – nevertheless areal borders need to be drawn correctly.

Addendum: having just now looked at the Munoz-Casteneda BICCN paper (on bioRxiv, although missing some key supplemental figures), it seems that in at least some if not most of the figures they correctly drew the MOp/SSp border, i.e., roughly perpendicular across the layers (Fig 2f, Fig 3b, among others). They also show in Fig 2c a comparison of three ways of drawing the border, which includes the border based on CCFv3, which one can see is quite different from the “expert manual annotation” version that seems to get it right. (An “algorithmically detected border” looks just as wrong as the CCFv3 based border.)

Response: The reviewer raised a concern that the border between MOp and SSp shown in the coronal sections in our paper, which is determined based on the latest Allen Mouse Common Coordinate Framework version 3 (CCF v3) (Ref 40, Wang et al, *Cell* 181, 936-953, 2020), seems to be against the well accepted knowledge that the cortical areal borders should be perpendicular to the cortical surface. All comments from this reviewer were based on this single concern. We would like to clarify that the MOp/SSp border, as well as all other cortical areal borders, in the CCF v3 are actually perpendicular to the cortical surface in the 3D space. The reason that these borders do not always appear perpendicular to the cortical surface line in coronal sections is because the coronal sections themselves are not always perpendicular to the cortical surface. In paracoronal sections that are perpendicular to the cortical surface, these borders are perpendicular to the cortical surface lines. A more detailed explanation is provided below.

The isocortex is a curved 3D sheet organized into layers where connections between the layers, as well as apical dendrites of excitatory neurons, are typically perpendicular to the surface in a columnar organization. However, the curvature makes it difficult to visualize along this perpendicular dimension. In CCF v3, a curved cortical coordinate system with streamlines was developed to approximate the perpendicular direction of the cortical sheet (Figure 1). By construction, the streamlines are perpendicular to the cortical surface and details of the streamlines computing are provided in Wang et al *Cell* 2020.

Figure 1: The average template brain of CCF v3 with the streamlines shown as colored lines.

In the way that cortical areas are parcellated in CCF v3, each entire streamline is assigned to a single cortical area, ensuring that the boundaries between cortical areas are necessarily perpendicular to the cortical surface in the 3D space. However, in a 2D section, the boundaries may not appear perpendicular to the cortical surface line depending on how the 2D plane is oriented. This is illustrated in Figure 2 below. In a coronal section that is not perpendicular to the cortical surface, the MOp/SSp border appears not perpendicular to the cortical surface line (Figure 2A, B). However, in a paracoronal section that is roughly perpendicular to the cortical surface at the MOp/SSp area (which is generated by raising the frontal end of the brain until the MOp/SSp part of cortical surface is roughly horizontal), the MOp/SSp border is perpendicular to the cortical surface line (Figure 2C, D). The same effect can be observed in other sections, for example the subsequent sections 0.4, 0.8 and 1.2 mm to the posterior

(Figure 3). (More details about this can be found at this blog generated by the Allen Institute: <https://community.brain-map.org/t/ccfv3-highlights-tilting-at-the-cortex/1000>).

Figure 2: Visualization of the MOp/SSp boundary in CCF v3 at a 2D coronal section not perpendicular to the cortical surface (A, B) and a paracoronal section perpendicular to the cortical surface at the MOp/SSp area. Borders between the brain structures are shown in blue and streamlines are shown in rainbow colors. (B) and (D) are zoom-in views of the boxed regions in the MOp/SSp area in (A) and (C), respectively. In (A, B), many streamlines cut through the coronal section, each appearing as a short segment, indicating that this section is not perpendicular to the cortical surface. Hence the MOp/SSp boundary also does not appear perpendicular to the cortical surface line in this section. In (C, D), much fewer streamlines cut through this paracoronal section and appear as much longer segments in the MOp/SSp area, indicating that this section is roughly perpendicular to the cortical surface in this area. Thus, in this section, the MOp/SSp boundary is perpendicular to the cortical surface line. The crosshair in all panels is centered on the same streamline (red).

Figure 3: Additional coronal (A-C) and paracoronal (D-F) sections showing the same effect as in Figure 2. Borders between the brain structures are shown in blue and streamlines are shown in rainbow colors. (A-

C) In the coronal sections that are not perpendicular to the cortical surface, the MOp/SSp boundaries do not appear perpendicular to the cortical surface lines. (D-F) In the corresponding paracoronal sections that are perpendicular to the cortical surface in the MOp/SSp area, the MOp/SSp boundaries are perpendicular to the cortical surface lines.

We also note that the Allen CCF is a reference atlas that has evolved over the years and the CCF v3 is the latest version (Wang et al *Cell* 2020). CCF v3 was created based on a new 3D reference brain: a volumetric average template generated from the average of 1,675 individual TissueCyte STPT imaged brains from the Allen Mouse Brain Connectivity Atlas (Oh et al *Nature* 508, 207-214, 2014). The entire average template brain was comprehensively parcellated and annotated directly in 3D space, with regions delineated based on registered multimodal reference datasets including histology stains, immunohistochemistry, transgene expression, in situ hybridization, and anterograde tracer connectivity experiments. While the brain areal boundaries may not be perfect in this CCF version and efforts in the community will continue to improve the accuracy of these boundaries, it is beyond the scope of our current paper to define a more precise MOp boundary than that defined in CCF v3. We used CCF v3 in our paper both because it is the latest version and because it is collectively decided by the BICCN for this consortium effort of MOp studies. The flagship paper and a number of other companion papers in this package use the CCF v3 as well, which provides consistency among multiple papers and allows analyses and interpretations of results in an integrated manner in the flagship paper in this package. Our MERFISH results in this companion paper, which provides spatially resolved single-cell transcriptomic data and hence the spatial locations of the molecularly identified cells, is a valuable resource independent of the MOp boundaries used; and as the accuracy of brain areal boundaries continues to improve over time, the results in our paper will continue to serve as a valuable resource to determine the cell type organization of the MOp and adjacent areas more accurately in the future.

Reviewer Reports on the Second Revision:

Referee #2:

Remarks to the Author:

I am also satisfied with the authors' responses to the reviewer regarding the border between primary motor cortex and primary somatosensory cortex and I recommend acceptance for publication.

Author Rebuttals to Second Revision:

There was no remaining comment from the reviewers to be addressed.

The final-round reviewer (Reviewer #2) was satisfied with the response that we made to address Reviewer 1's comments from the previous round. In line with our response, we have added an explanation to clarify why the brain areal borders do not always appear perpendicular to the cortical surface line in coronal sections, whereas these areal borders are always perpendicular to the cortical surface in 3D space in Allen CCF v3. This explanation is added in the caption of Extended Data Figure 2 where an Allen CCF v3 brain atlas image is used.